# Enhancement of erythropoietic output by Cas9-mediated insertion of a natural variant in haematopoietic stem and progenitor cells

Sofia E. Luna [1,2,8], Joab Camarena [1,2,8], Jessica P. Hampton [1,2],
Kiran R. Majeti[1,2], Carsten T. Charlesworth[1,2], Eric Soupene[3], Sridhar Selvaraj[1,2],
Kun Jia[4,5,6], Vivien A. Sheehan [7], M. Kyle Cromer [4,5,6] & 
Matthew H. Porteus [1,2]

Some gene polymorphisms can lead to monogenic diseases, whereas other polymorphisms may confer beneficial traits. A well-characterized example is congenital erythrocytosis—the non-pathogenic hyper-production of red blood cells—that is caused by a truncated erythropoietin receptor. Here we show that Cas9-mediated genome editing in CD34⁺ human haematopoietic stem and progenitor cells (HSPCs) can recreate the truncated form of the erythropoietin receptor, leading to substantial increases in erythropoietic output. We also show that combining the expression of the cDNA of a truncated erythropoietin receptor with a previously reported genome-editing strategy to fully replace the *HBA1* gene with an *HBB* transgene in HSPCs (to restore normal haemoglobin production in cells with a β-thalassaemia phenotype) gives the edited HSPCs and the healthy red blood cell phenotype a proliferative advantage. Combining knowledge of human genetics with precise genome editing to insert natural human variants into therapeutic cells may facilitate safer and more effective genome-editing therapies for patients with genetic diseases.

Many of the initial applications of clinical genome editing have aimed to correct or compensate for disease-causing mutations of monogenic diseases. However, human genetic variation is more nuanced than monogenic diseases as there are also variants that appear to confer positive health benefits. For example, people with bi-allelic deletions in *CCR5* show resistance to HIV infection, a variety of polymorphisms cause upregulation of foetal haemoglobin and mutations in *PCSK9* cause low cholesterol levels[1–3]. It is important, however, to broadly assess variations that occur in only small numbers of people both for their potential risks and for their potential benefits.

Congenital erythrocytosis (CE) is a rare phenotype in which people have higher than normal levels of red blood cells (RBCs) and consequently elevated haemoglobin. Although there are multiple genetic variants that can lead to this condition, perhaps the best-characterized genotype was first identified in the family of a Finnish Olympic-gold-medal-winning cross-country skier who was found to have levels of haemoglobin >50% higher than normal[4]. This elevated haemoglobin was attributed to truncations in the erythropoietin receptor (tEPOR) in which the intracellular inhibitory domain to erythropoietin (EPO) signalling is eliminated[4,5]. This domain

¹Department of Pediatrics, Stanford University, Stanford, CA, USA. ²Institute for Stem Cell Biology and Regenerative Medicine, Stanford University, Stanford, CA, USA. ³Department of Pediatrics, University of California, San Francisco, Oakland, CA, USA. ⁴Department of Surgery, University of California San Francisco, San Francisco, CA, USA. ⁵Department of Bioengineering and Therapeutic Sciences, University of California San Francisco, San Francisco, CA, USA. ⁶Eli and Edythe Broad Center for Regeneration Medicine, University of California San Francisco, San Francisco, CA, USA. ⁷Aflac Cancer and Blood Disorders Center, Department of Pediatrics, Emory University School of Medicine, Atlanta, GA, USA. ⁸These authors contributed equally: Sofia E. Luna, Joab Camarena. ✉e-mail: kyle.cromer@ucsf.edu; mporteus@stanford.edu

contains binding sites for SHP1 that normally leads to downregulation of EPO-dependent JAK2–STAT5 signalling (Supplementary Fig. 1a). Further studies have shown that tEPOR does not create a constitutively active EPOR signalling cascade but rather imparts hypersensitivity to EPO[5,6]. As a consequence, these kindreds with tEPOR typically present with abnormally low levels of EPO, indicating a new homoeostasis is attained to prevent CE from becoming pathogenic. There have been reports of thrombotic and haemorrhagic events likely due to erythrocytosis but many of these events have a benign clinical course[7]. More importantly, families with CE have not shown an increased predisposition to cancer, showing that this is not a premalignant genetic condition[8].

Although previous studies have investigated the effects of viral-mediated delivery and expression of tEPOR[9–11], random insertion into the genomes of billions of haematopoietic stem and progenitor cells (HSPCs) in the context of bone marrow (BM) transplant presents a serious safety concern and has resulted in a 'black box' warning in the United States for lovotibeglogene autotemcel, a lentiviral gene therapy drug approved for sickle cell disease (SCD)[12]. In addition, such instances of viral-mediated delivery require expression of tEPOR using a non-native exogenous promoter, which departs from native EPOR regulation and has the potential for unintended consequences, such as pathogenic polycythaemia. Nonetheless, viral-mediated expression studies provide a proof of concept that shows that tEPOR expression can lend a selective advantage to transduced cells and provide a foundation for the utilization of more advanced genome-engineering modalities.

Genome editing is a powerful method that enables the precise changing of nucleotides in the DNA of a cell. There are multiple genome-editing strategies, including nuclease-based insertion–deletion (indel) formation, base editing and prime editing, but the most versatile approach to genome editing is homology-directed repair (HDR). In HDR, a nuclease-induced double-strand break (DSB) is repaired using a donor template. The natural donor template for HDR is the sister chromatid and the natural repair pathway is homologous recombination. By providing a donor template that resembles a sister chromatid with large homology arms flanking the intended cut site, the homologous recombination machinery can use this 'substitute' sister chromatid to repair the DSB. HDR editing is the most flexible approach because it can create single nucleotide changes, precisely insert large gene cassettes and even swap out large genomic regions for other gene sequences[13–16]. We use all these applications of HDR in this work, including direct creation of the naturally occurring variant found in a human kindred.

One of the challenges in haematopoietic stem cell gene therapy is to achieve sufficient engraftment of the genetically engineered cells to have a beneficial clinical effect without increasing risk. To make this possible, effort is exerted to maximize editing frequencies in HSPCs[17–19]. Even if clinically relevant editing frequencies are achieved, high-morbidity chemotherapeutic regimens are currently required to create niche space in the BM for these edited HSPCs, which can create toxicities, including oncogenic risk[20–22]. In this work we aimed to give edited cells a selective advantage such that low levels of engraftment might still result in a clinical benefit, perhaps enabling less toxic conditioning, through the use of genome editing to recreate the CE phenotype by engineering tEPOR into HSPCs in different ways. We find that when tEPOR is engineered into human HSPCs using genome editing, there is a substantial selective advantage to the derived RBCs. We then show that this selective advantage can be coupled to a therapeutic gene edit to give the cells with the therapeutic edit a selective advantage in RBC development, without affecting the stem and progenitor cells. In this way, we show the power of combining human genetics with precision genome editing to potentially enable safer and more effective genome-editing therapies for patients with serious genetic diseases, particularly those involving RBCs.

## Results

### Cas9-guided *EPOR* truncation enhances erythroid proliferation

Truncating mutations in the *EPOR* gene that cause clinically benign CE[4,8] provide a potentially safe avenue to increase erythropoietic output from genome-edited HSPCs. In this study we designed Cas9 single-guide RNAs (sgRNAs)[17] (termed *EPOR*-sg1 and *EPOR*-sg2) that overlap the location of the originally identified nonsense mutation, *EPOR* c.1316G>A (p.W439X; Mäntyranta variant)[4] (Fig. 1a and Supplementary Fig. 2a). Our hypothesis was that targeting this site in exon 8 with Cas9 would create a spectrum of indels, a subset of which would result in a frameshift of the reading frame and yield premature downstream stop codons in the *EPOR* gene.

To test this hypothesis, we precomplexed each sgRNA with high-fidelity Cas9 protein[23] and delivered these ribonucleoprotein (RNP) complexes to human CD34[+] HSPCs. At 2–3 days postediting, we transferred cells into culture medium that promotes erythroid differentiation over the course of 2 weeks (Fig. 1a)[24]. To determine whether edited HSPCs have a proliferative advantage compared with unedited cells, we collected genomic DNA at day 0, 4, 7, 11 and 14 of RBC differentiation. We then quantified indel frequency by polymerase chain reaction (PCR) amplification followed by Sanger sequencing and decomposition analysis using TIDE[25]. In the absence of a selective advantage or disadvantage, the percentage of edited alleles in cells at the beginning and end of RBC differentiation will be roughly equivalent—which is what we observe for the editing frequency of the *HBB* sgRNA used for correction of SCD. However, we observe that the editing frequency of *EPOR*-targeting sgRNAs increases significantly over the course of erythroid differentiation, to a greater extent in *EPOR*-sg1 than in *EPOR*-sg2 (*P* = 0.0016 for *EPOR*-sg1 from day 0 to day 14; Fig. 1b and Supplementary Fig. 2b). In addition, we show that the increase in indels for both *EPOR*-sg1 and *EPOR*-sg2 is predominantly driven by indels that yield downstream stop codons (Extended Data Fig. 1a,b and Supplementary Fig. 2c,d). This indicates that edited cells, particularly those with premature stop codons, are outcompeting unedited cells because of the EPO hypersensitivity of *tEPOR*-expressing cells in culture[5,6].

As not all indels created by the sgRNAs cause truncations in *EPOR*, we speculated that we could increase this proliferative effect by using HDR to insert a stop codon at the exact location of the original variant (c.1316G>A). To accomplish this, we designed an adeno-associated virus serotype 6 (AAV6) repair template vector that introduces a stop codon into *EPOR* at the 439th amino acid (W439X) followed by a *BGH*-poly(A) tail to terminate transcription. We also included a downstream GFP marker driven by the constitutive human *UbC* promoter to ensure that each GFP[+] allele harbours the intended *EPOR*-truncating mutation. The entire integration cassette was flanked by 950 bp homology arms that corresponded to the genomic DNA immediately upstream and downstream of the intended Cas9 cut site created by the more effective *EPOR*-sg1 (Fig. 1c). To determine whether this editing strategy was also able to drive enrichment of genome-edited RBCs, we complexed *EPOR*-sg1 with Cas9 protein and delivered this by electroporation to human CD34[+] HSPCs followed by transduction with an AAV6 DNA repair template. At 2–3 days postediting, we either maintained cells in HSPC media or began erythroid differentiation with 3 U ml[−1] of EPO (+EPO), as has been previously described[14], or with 0 U ml[−1] of EPO (−EPO) to determine whether *tEPOR*-expressing cells retain EPO sensitivity or became EPO independent during their differentiation. At day 14 of erythroid differentiation, we stained for established RBC markers[14] and analysed cells using flow cytometry. We observed no differentiation when cells were kept in HSPC media and efficient RBC differentiation in all treatments with EPO. In the edited conditions in the absence of EPO, we observed moderate differentiation that may indicate the hypersensitivity of *tEPOR*-expressing cells to trace amounts of EPO in the media, as has been previously observed

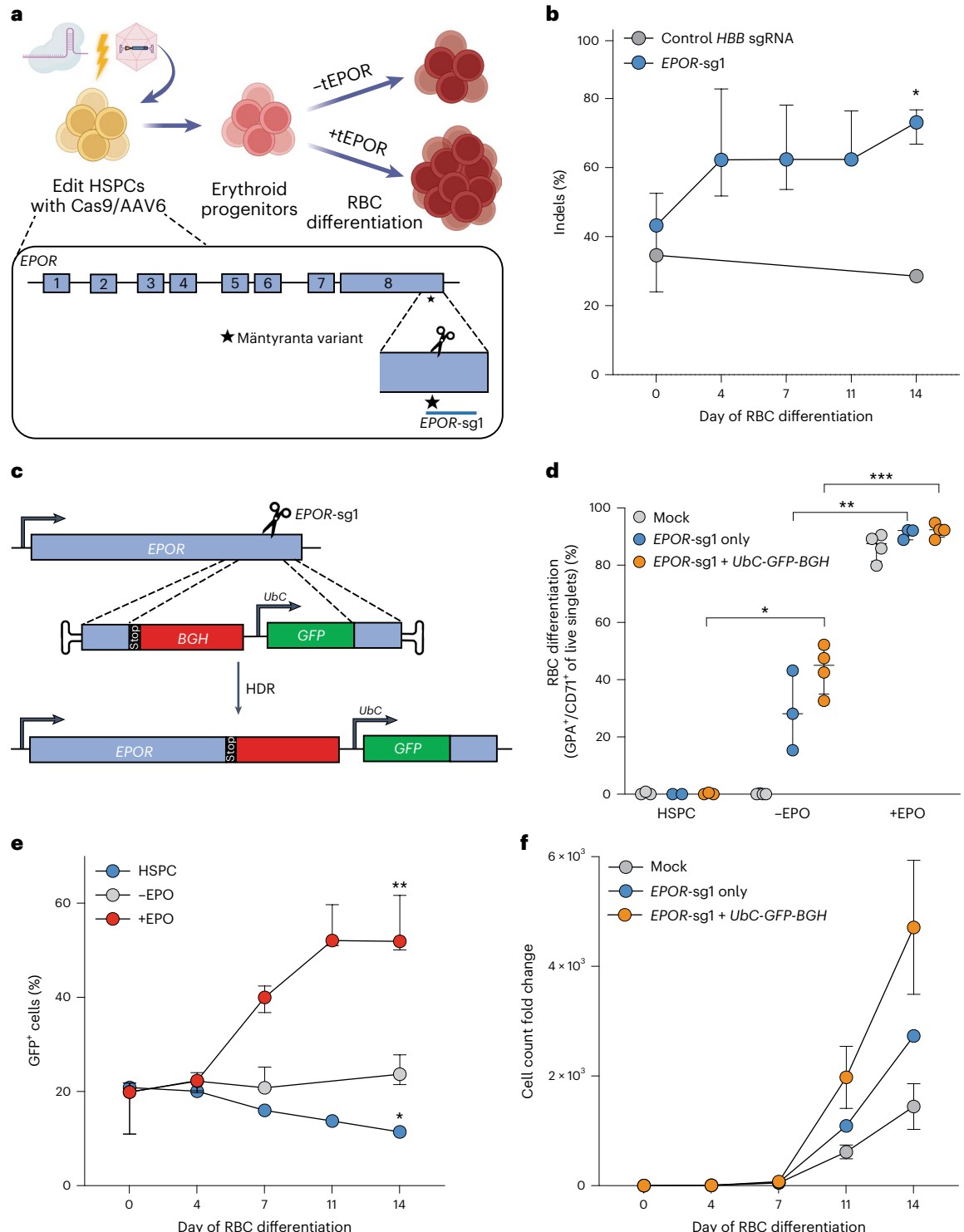

**Fig. 1 | Cas9-guided *EPOR* truncation in HSPCs enhances erythroid proliferation. a**, Schematic of HSPC editing and model of *tEPOR*'s effect. Representation of *EPOR* gene and location of the candidate sgRNA (*EPOR*-sg1) indicated by a line. Location of c.1316G>A mutation is denoted by the star. Created with BioRender.com. **b**, Frequency of indels created by *EPOR*-sg1 in primary human CD34+ HSPCs over the course of erythroid differentiation compared with control *HBB* sgRNA. Points represent median ± interquartile range. Values represent biologically independent HSPC donors: $n = 5$ for *EPOR*-sg1 and $n = 1$ for control *HBB* sgRNA. *$P = 0.0016$ of day 0 versus day 14 by unpaired two-tailed *t*-test. **c**, Genome-editing strategy when using an AAV6 DNA repair template to introduce the *EPOR* c.1316G>A mutation followed by a *BGH*-poly(A) region and *UbC*-driven *GFP* reporter. **d**, Percentage of GPA+/CD71+ of live single cells on day 14 of differentiation. Bars represent median ± interquartile range. Values represent

biologically independent HSPC donors: $n = 2$–$3$ for HSPC and $n = 3$–$4$ for −EPO and +EPO conditions. *$P = 0.0016$ of −EPO versus HSPC conditions; **$P = 0.003$, ***$P = 0.0001$ of −EPO versus +EPO conditions by unpaired two-tailed *t*-test. **e**, Percentage of GFP+ cells of live single cells maintained in RBC media with or without EPO or HSPC media as determined by flow cytometry. Points represent median ± interquartile range. Values represent biologically independent HSPC donors: $n = 2$ for HSPC condition and $n = 3$–$4$ for −EPO and +EPO conditions. *$P = 0.04$, **$P = 0.0006$ of day 0 versus day 14 by unpaired two-tailed *t*-test. **f**, Fold change in cell count throughout RBC differentiation (for example, if at day 0 starting cell numbers were $1 \times 10^5$ cells total, then a fold count change of 1,000 would yield a total cell number of $1 \times 10^8$ at day 14). Points represent mean ± s.e.m. Values represent biologically independent HSPC donors: $n = 3$ for mock and *EPOR*-sg1 + *BGH* and $n = 2$ for *EPOR*-sg1.

(Fig. 1d)[4]. By analysing GFP[+] cells over the course of RBC differentiation, in the *EPOR*[W439X]-edited conditions we observed a significant increase in the frequency of edited cells in only the +EPO conditions ($P = 0.0006$ comparing day 0 to day 14; Fig. 1e). At the end of differentiation, by looking at both GFP[+] cells and frequency of indel formation, we estimated that almost all the RBCs are derived from cells with a truncated EPOR due to either an indel or a UbC-GFP knock-in event (Extended Data Fig. 2a). In addition to the competitive advantage that *tEPOR* expression gives to edited cells over the course of RBC differentiation, we also observed increased RBC production in both *EPOR*-sg1 and *EPOR*-sg1 + *UbC-GFP-BGH* conditions compared with mock control in the +EPO condition (average $1.45 \times 10^3$ total fold increase in mock-edited cells versus $4.71 \times 10^3$ in *EPOR*-sg1 + *UbC-GFP-BGH* over the 14-day RBC differentiation; Fig. 1f). These increased cell counts were not observed in the −EPO condition or when cells were maintained in HSPC media, indicating an EPO-driven increase in erythroid proliferation in cells expressing *tEPOR* (Extended Data Fig. 2b). We also assessed whether a gradient of EPO concentrations (0–20 U ml[−1]) over the course of RBC differentiation yielded varying degrees of enrichment of edited cells (Extended Data Fig. 3a). On day 14, when compared with −EPO, we found minimal differences in GFP[+] cells in the 1 U ml[−1], 3 U ml[−1] and 20 U ml[−1] conditions and only a minor reduction in GFP[+] cells in the 0.3 U ml[−1] EPO condition (Extended Data Fig. 3b,c), indicating that even low levels of EPO are sufficient to impart a selective advantage to edited cells, consistent with previous work on the natural variants[5,6]. We also observed comparable levels of RBC differentiation at all concentrations except −EPO (Extended Data Fig. 3d).

In terms of the safety of this editing strategy, we show that introduction of these *EPOR* indels yields RBCs with production of both foetal and adult haemoglobin tetramers following haemoglobin tetramer high-performance liquid chromatography (HPLC) (Supplementary Fig. 3a). We also found similar colony number and lineage distribution from CD34[+] HSPCs plated into wells containing methylcellulose media either with or without EPO that were scored for colony-formation ability after 14 days (Supplementary Fig. 3b). As expected, there was a marked decrease in the ability to form erythroid burst-forming unit (BFU-E) colonies in the absence of EPO even if the cells contained the tEPOR. These data reinforce the idea that truncation of EPOR does not alter the lineage bias of HSPCs but rather has an effect only after commitment to the erythroid lineage. In addition, although transient delivery of high-fidelity Cas9 has been shown to be highly specific to the on-target site[26], we also evaluated potential off-target effects of the *EPOR*-sg1−RNP complex in HSPCs. We found that 94% (73 of 78) of candidate off-target sites with scores previously shown to be most informative for identifying sites with potential off-target activity[27] resided in intergenic or intronic regions of the genome (Extended Data Fig. 4a,b). We further interrogated potential off-target activity at the five sites that resided in exonic or untranslated regions (UTRs) of genes (Extended Data Fig. 4c) and found no evidence of off-target

activity in *EPOR*-sg1-edited cells when compared with mock-edited cells (Extended Data Fig. 4d,e).

## *tEPOR* at a safe-harbour locus replicates proliferative effect

Given the therapeutic utility of transgene integration at safe-harbour sites[28], we hypothesized that integration of a *tEPOR* cDNA at a safe-harbour site may also enable increased erythroid production from edited HSPCs while leaving the endogenous *EPOR* locus intact. Given the fact that integration at the *CCR5* locus is an established method for delivery of therapeutic transgenes in HSPCs[29], we developed a custom AAV6-packaged DNA repair template that would facilitate integration of an exogenous human *UbC* promoter driving expression of *tEPOR* cDNA followed by a *T2A-YFP-BGH* reporter (Fig. 2a). Given the strong, constitutive expression of the *UbC* promoter, this method of insertion is expected to express *tEPOR* ubiquitously in all haematopoietic cell types, regardless of lineage.

To test this strategy for *tEPOR* expression, we edited HSPCs with Cas9 complexed with an established sgRNA targeting exon 2 of *CCR5* (*CCR5*-sg3)[29], immediately followed by transduction with our custom DNA repair template. We then performed RBC differentiation postediting and analysed the kinetics of editing frequency, YFP expression and erythroid differentiation using droplet digital PCR (ddPCR) and flow cytometry. As with endogenous *EPOR* truncation strategies, we observed more efficient erythroid differentiation in all treatments with EPO compared with the −EPO conditions (Fig. 2b). Although flow cytometry confirmed the ubiquitous expression of YFP in edited cells, regardless of presence of CD71 and GPA erythroid markers during differentiation (Fig. 2c), we did observe significant enrichment of YFP-expressing RBCs in the presence of EPO ($P < 0.0001$ when comparing day 0 with day 14 of RBC differentiation; Fig. 2d). This enrichment was confirmed at the genomic level by ddPCR that showed an increase in the percentage of edited alleles when *tEPOR*-expressing cells were subjected to erythroid differentiation, increasing by an average of 7.3-fold in the presence of EPO, and enrichment to a limited degree in the −EPO condition (Extended Data Fig. 5a). With this editing strategy, we observed increased RBC production in the *CCR5*-sg3 + *tEPOR* condition and no increase in proliferation of *CCR5*-sg3 alone compared with mock control cells in the +EPO condition. We saw no increased proliferation in the cells maintained in HSPC media or in the −EPO condition (Extended Data Fig. 5b). We then assessed if enrichment of edited cells differed along a gradient of EPO levels during RBC differentiation and again found that there were minimal differences in YFP[+] cells in the 1 U ml[−1], 3 U ml[−1] and 20 U ml[−1] conditions and a minor decrease in the 0.3 U ml[−1] EPO condition compared with that in the −EPO condition (Extended Data Fig. 5c,d). There were comparable levels of RBC differentiation in all conditions except that of −EPO (Extended Data Fig. 5e).

We again observed that edited and unedited cells had no noticeable difference in lineage commitment or colony-forming ability following a colony-forming unit (CFU) assay (Supplementary Fig. 4a).

**Fig. 2 | Integration of *tEPOR* cDNA shows an erythroid-specific proliferative effect. a**, Genome-editing strategy to introduce *tEPOR-T2A-YFP-BGH*-poly(A) cDNA at the *CCR5* locus with expression driven by a ubiquitous *UbC* promoter. **b**, Percentage of GPA[+]/CD71[+] of live single cells on day 14 of differentiation following introduction of *tEPOR* at the *CCR5* locus. Bars represent median ± interquartile range. Values represent biologically independent HSPC donors: $n = 2$–$3$ for HSPC condition and $n = 2$–$4$ for −EPO and +EPO conditions. *$P = 0.0018$ for −EPO versus HSPC conditions, **$P < 0.0001$ for −EPO versus +EPO conditions by unpaired two-tailed *t*-test. **c**, Representative flow cytometry plots of one donor of *CCR5*-sg3 + *tEPOR*-edited HSPCs on day 14 of RBC differentiation in the +EPO condition. **d**, Percentage of YFP[+] cells of live single cells as determined by flow cytometry. Points represent mean ± s.e.m. Values represent biologically independent HSPC donors: $n = 2$ for HSPC condition, $n = 3$ for −EPO condition and $n = 3$–$4$ for +EPO condition. *$P = 0.0003$ for day 0 versus day 14 by unpaired

two-tailed *t*-test. **e**, Genome-editing strategy to introduce *tEPOR-T2A-YFP* cDNA at the *HBA1* locus by whole gene replacement to place integration cassette under regulation of the endogenous *HBA1* promoter. **f**, Percentage of GPA[+]/CD71[+] of live single cells on day 14 of differentiation following introduction of *tEPOR* cDNA at the *HBA1* locus. Bars represent median ± 95% confidence interval. Values represent biologically independent HSPC donors: $n = 2$ for HSPC condition and $n = 2$–$3$ for −EPO and +EPO condition. *$P = 0.0002$ for −EPO to +EPO condition by unpaired two-tailed *t*-test. **g**, Representative flow cytometry plots of one donor of *HBA1*-sg4 + *tEPOR*-edited HSPCs on day 11 of RBC differentiation in the +EPO condition. **h**, Percentage of YFP[+] cells of live single cells as determined by flow cytometry. Points represent mean ± s.e.m. Values represent biologically independent HSPC donors: $n = 2$ for HSPC condition and $n = 3$ for −EPO and +EPO condition. FSC-A, forward scatter area; FITC, fluorescein isothiocyanate.

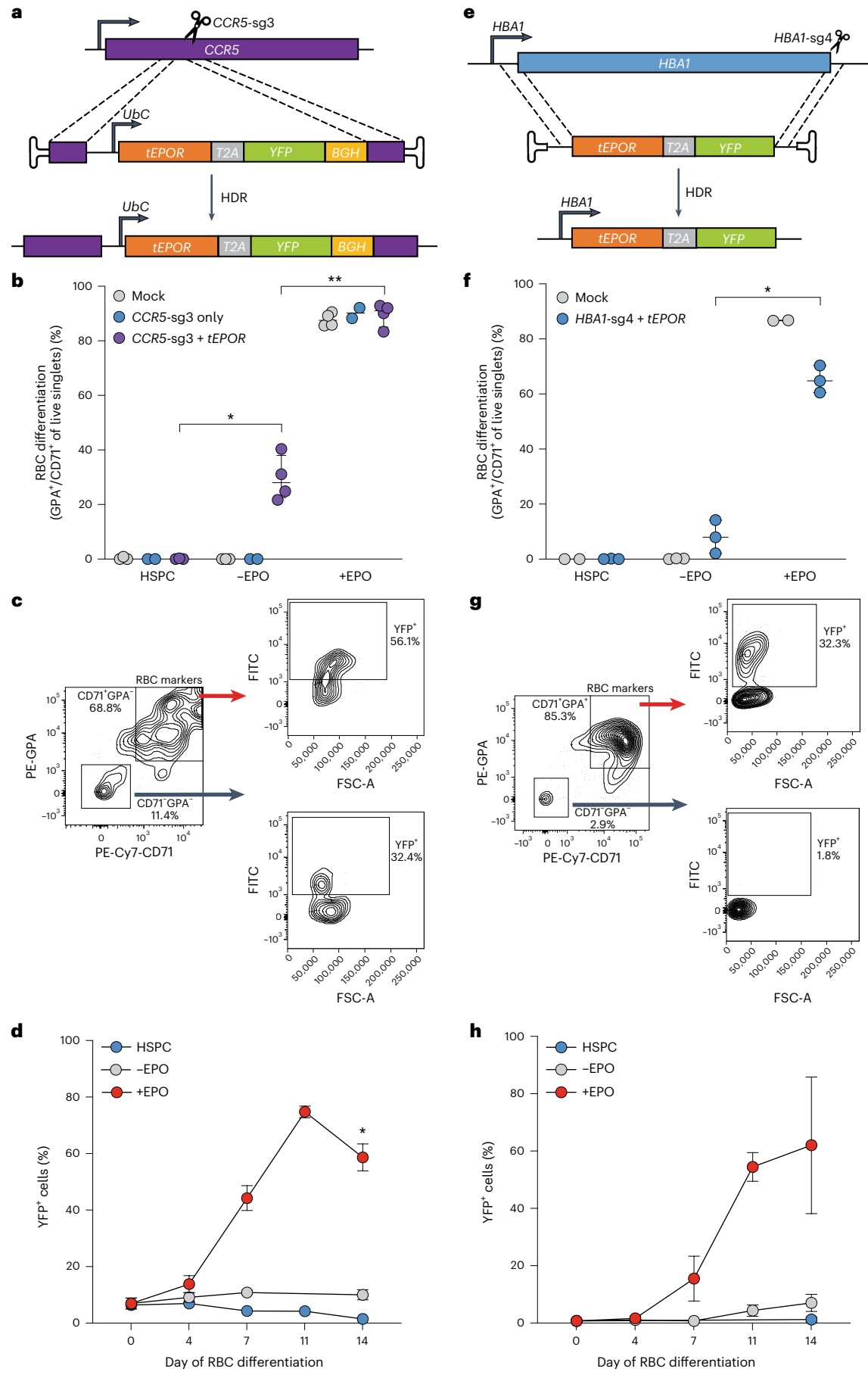

In addition, we analysed RBC postdifferentiation using haemoglobin tetramer HPLC and found that *UbC*-mediated expression of *tEPOR* resulted in both HgbF and HgbA expression but a relative increase in HgbF expression (Supplementary Fig. 4b). These results indicate that expression of *tEPOR* from a safe-harbour site is an effective means of driving increased RBC production from genome-edited HSPCs.

### *tEPOR* at *HBA1* shows erythroid-specific proliferation

Although integration at a safe-harbour locus effectively increased erythropoietic output from edited HSPCs, there is the concern that constitutive expression in all cell types could disrupt stemness or lead to other unintended effects. As an alternative, we can introduce promoterless transgenes into endogenous genes for the integration cassette to be regulated by endogenous expression machinery. For instance, in previous work we designed a genome-editing strategy to fully replace the *HBA1* gene with an *HBB* transgene to correct β-thalassaemia[14]. We found that because α-globin is produced by duplicate genes, *HBA1* may serve as a safe-harbour site to deliver custom payloads with strong erythroid-specific expression.

We therefore hypothesized that integration of the *tEPOR* cDNA at the *HBA1* locus could further enhance production of edited RBCs while avoiding potential complications with ubiquitous transgene expression because of the specificity of expression of *HBA1* in the RBC lineage. To test this hypothesis, we designed a custom integration cassette (also packaged in AAV6) to use with an established sgRNA (*HBA1*-sg4) to introduce a promoterless *tEPOR* cDNA followed by a *T2A-YFP* reporter under expression of endogenous *HBA1* regulatory machinery (Fig. 2e). Following editing, we observed efficient erythroid differentiation in all treatments with EPO and little to no differentiation in the absence of EPO (Fig. 2f). We found that this integration strategy indeed yielded RBC-specific expression of YFP, which was only detectable as cells gained CD71 and GPA erythroid markers (Fig. 2g). Over the course of RBC differentiation, we observed dramatic enrichment of YFP+ cells exclusively in the +EPO condition (Fig. 2h). Although we observed a mild degree of enrichment of edited alleles using ddPCR in the absence of EPO, this effect was more pronounced in the presence of EPO, eliciting an average 4.4-fold increase in the percentage of edited alleles over the course of RBC differentiation (Extended Data Fig. 6a). We again cultured edited cells in a gradient of EPO during RBC differentiation and observed minimal differences in YFP+ cell enrichment in all conditions containing EPO (Extended Data Fig. 6b,c). There were also comparable levels of RBC differentiation in all conditions except that of −EPO (Extended Data Fig. 6d).

Again we found that cells edited with *tEPOR* at *HBA1* showed similar colony-forming ability as mock-edited cells (Supplementary Fig. 5a). In addition, we found that *tEPOR*-expressing cells produce ratios of human haemoglobin that are similar to unedited RBCs (Supplementary Fig. 5b).

### *HBB-tEPOR* increases production of thalassaemia-corrected RBCs

As the above results show that *tEPOR* expression yields increased RBC production from edited HSPCs, we then sought to couple this selective advantage with a therapeutic gene edit. We chose to combine tEPOR with our previous β-thalassaemia correction approach to simultaneously correct the disease and increase production of clinically meaningful RBCs from these corrected HSPCs. One way this can be accomplished is by creating a bicistronic cassette that links expression of the therapeutic full-length *HBB* transgene with a *tEPOR* cDNA. Previous studies have found that the type of linker domain used can have a great bearing on transgene expression and protein function—particularly when function is dependent on formation of protein complexes, as is the case with the globin genes[14]. Therefore, we designed and tested a variety of AAV6 repair template vectors linking the two genes using standard T2A peptides, optimized T2A peptides with furin cleavage

sites[30] (referred to as FuT2A) and internal ribosome entry sites (referred to as IRES) and by driving *tEPOR* expression from a separate exogenous promoter—human *PGK1* (referred to as PGK). To evaluate the different vectors, we edited healthy donor HSPCs as previously described at *HBA1* using the bicistronic AAV6 repair templates and evaluated their ability to differentiate and enrich for edited RBCs over the course of erythroid differentiation (Fig. 3a). Although efficient RBC differentiation was achieved in all editing conditions (Fig. 3b), we found that all four bicistronic cassettes drove >2-fold enrichment of edited alleles (range, 2.1-fold to 3.5-fold; *P* = 0.003 for *PGK-tEPOR*, *P* = 0.0055 for *tEPOR-T2A*, *P* = 0.0259 for *tEPOR-FuT2A* and *P* = 0.0003 for *IRES-tEPOR* when comparing day 0 to day 14; Fig. 3c,d). We note that allele-targeting frequencies of 60% translate into >80% of the cells having at least one allele targeted (cell-targeting frequency) and thus we would not expect to see much more enrichment than we observed.

To ensure this strategy was also effective in patient-derived cells, we tested these bicistronic vectors in HSPCs derived from patients with SCD, this time comparing them with a therapeutic full-length *HBB* transgene[14]. All the constructs are knocked in to the *HBA1* locus without disrupting the endogenous *HBB* locus expressing HgbS. We found that vectors did not disrupt erythroid differentiation when compared with mock-edited cells in the same donor, although their ability to differentiate was likely impacted by the variable quality of the frozen patient samples (Extended Data Fig. 7a). Again, we observed >2-fold enrichment of edited alleles for all four bicistronic vectors (range, 2.0-fold to 3.1-fold), but no change in editing frequency for the original β-thalassaemia correction vector (*P* = 0.0061 for *PGK-tEPOR*, *P* = 0.011 for *tEPOR-T2A*, *P* = 0.0016 for *tEPOR-FuT2A*, *P* = 0.0153 for *IRES-tEPOR* when comparing day 0 with day 14; Fig. 3e,f). We estimate that at the end of the differentiation almost all the cells have at least one allele with the *HBB-tEPOR* knock-in and thus no biological drive for further enrichment. When we analysed differentiated RBCs for haemoglobin tetramers by HPLC, we found that the *T2A* vectors showed almost no HgbA expression (consistent with a previous observation that addition of a *T2A* can disrupt HBB protein function[31]). In contrast, we found that *PGK-tEPOR* and *IRES-tEPOR* vectors showed an improvement in HgbA production relative to the *HBB*-only edited cells (Extended Data Fig. 7b). There was no change in the HgbF expression in these samples.

### Multiplexed *tEPOR* and *HBB* editing increases *β-globin* mRNA

In lieu of coupling the *tEPOR* cDNA and therapeutic edit at the same locus, an alternative strategy would be to multiplex two editing events at different loci to simultaneously truncate the endogenous *EPOR* and introduce the original β-thalassaemia correction vector at *HBA1*. This strategy may have the additional advantage that the endogenous *EPOR* truncation will more reliably recapitulate CE. We hypothesized that we could simultaneously deliver Cas9 separately precomplexed with *EPOR*-truncating *EPOR*-sg1 and *HBA1*-sg4 gRNA and then transduce HSPCs with both the β-thalassaemia correction vector and the W439X *EPOR*-targeting vector. As homology arms of each vector are specific for each site—*HBB* for the *HBA1* locus and W439X for the *EPOR* locus—integration for each vector will occur only at the intended locus. This strategy may allow simultaneous correction of β-thalassaemia and increased erythropoietic output from corrected HSPCs. For this to be maximally effective, the two editing events must be present in the same cell. Therefore, during editing we used a DNA-PKcs inhibitor to increase the frequency of template integration at each locus[32,33]. Following editing, we determined whether this multiplexed editing strategy increases the frequency of corrected RBCs over the course of erythroid differentiation.

To model the clinical setting where both edited and unedited HSPCs would occupy the patient's BM, we introduced unedited cells at various concentrations at the start of erythroid differentiation (Fig. 4a). Importantly, none of the multiplexed conditions disrupted erythroid differentiation compared with single-edited *HBB* or mock

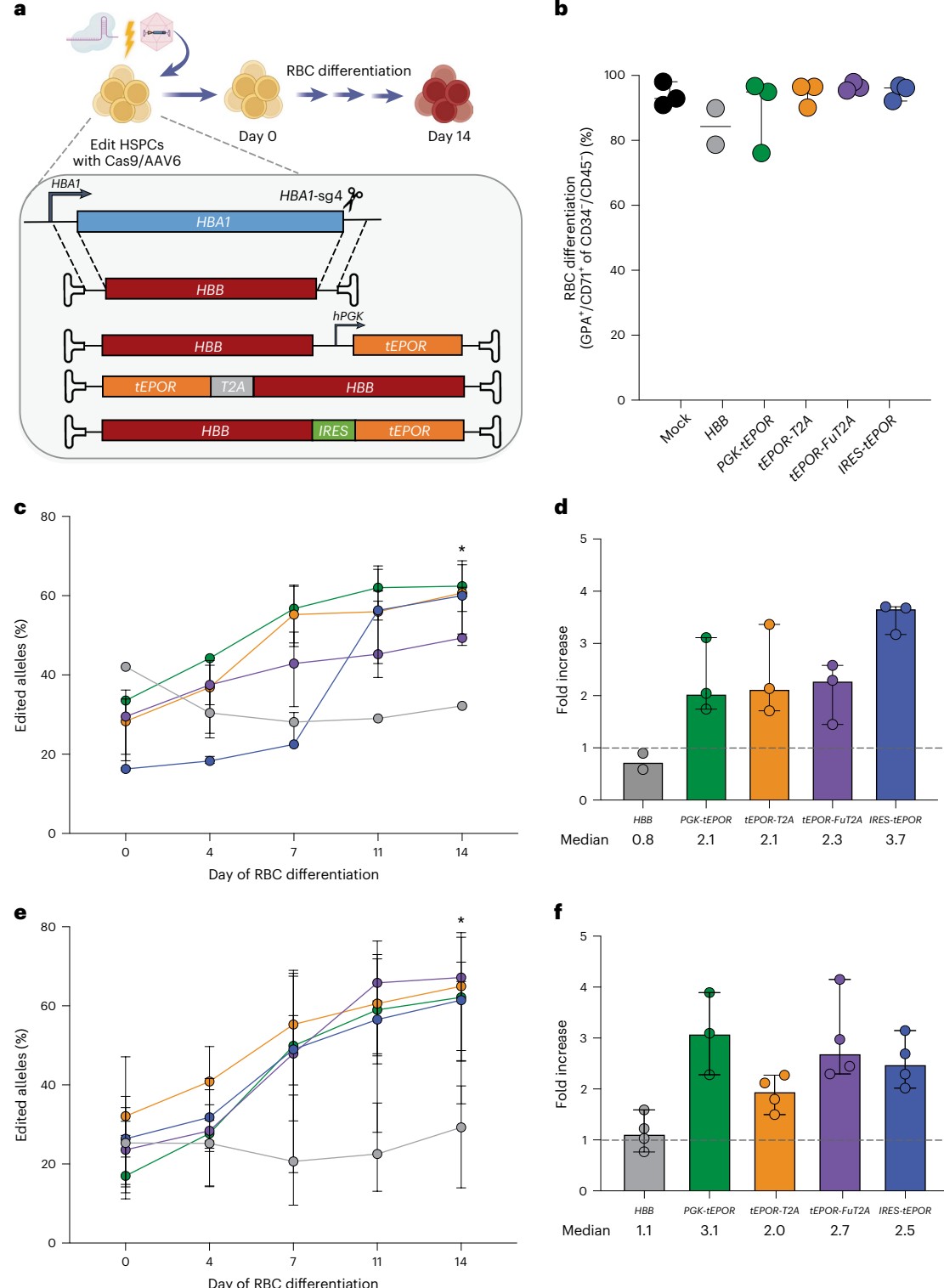

**Fig. 3 | Therapeutic editing frequencies are achieved using bicistronic *HBB-tEPOR* cassette. a**, Design of *HBB* (control) and *HBB-tEPOR* (bicistronic) AAV6 donor cassettes targeted to the *HBA1* locus by whole gene replacement. **b**, Percentage of GPA⁺/CD71⁺ of CD34⁻/CD45⁻ cells on day 14 as determined by flow cytometry. Points are shown as median ± 95% confidence interval. Values represent biologically independent HSPC donors: *n* = 2 for *HBB* and *n* = 3 for all other vectors. **c**, Percentage of edited alleles for control (*HBB*) and bicistronic *HBB-tEPOR* in cord-blood-derived CD34⁺ cells over the course of RBC differentiation. Points are shown as median ± 95% confidence interval. Values represent biologically independent HSPC donors: *n* = 2 for *HBB*, *n* = 3 for all other vectors. *\*P* = 0.003 for *PGK-tEPOR*, *P* = 0.0055 for *tEPOR-T2A*, *P* = 0.0259 for *tEPOR-FuT2A*, *P* = 0.0003 for *IRES-tEPOR* (day 0 versus day 14) by unpaired

two-tailed *t*-test. **d**, Fold change in edited alleles from the beginning (day 0) to end (day 14) of RBC differentiation. The dashed line represents no fold change. Bars represent median ± 95% confidence interval. **e**, Percentage of edited alleles for control (*HBB*) and bicistronic *HBB-tEPOR* vectors in cells from patients with SCD over the course of RBC differentiation. Points are shown as median ± 95% confidence interval. Values represent biologically independent HSPC donors: *n* = 3 for *PGK-tEPOR* and *n* = 4 for all other vectors. *\*P* = 0.0061 for *PGK-tEPOR*, *\*P* = 0.011 for *tEPOR-T2A*, *\*P* = 0.0016 for *tEPOR-FuT2A*, *\*P* = 0.0153 for *IRES-tEPOR* (day 0 versus day 14) by unpaired two-tailed *t*-test. **f**, Fold change in edited alleles from the beginning (day 0) to end (day 14) of RBC differentiation. The dashed line represents no fold change. Bars represent median ± 95% confidence interval.

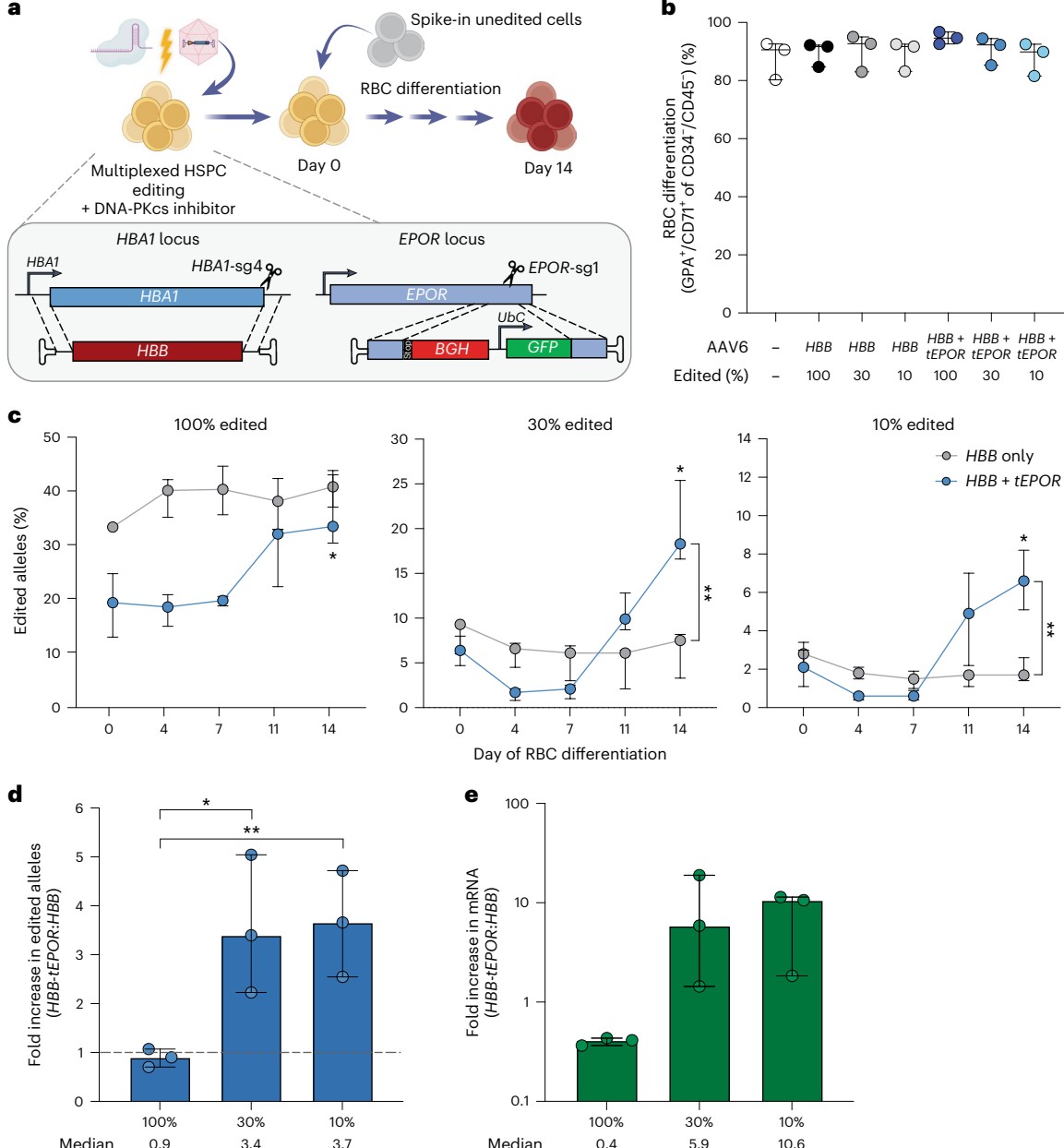

**Fig. 4 | Multiplexed editing of *EPOR* and *HBA1* leads to robust increase in *HBB* mRNA within editing HSPCs. a**, Schematic of multiplexed editing strategy with spike-in of unedited cells at the start of erythroid differentiation to model HSPC transplantation. **b**, Percentage of GPA⁺/CD71⁺ of CD34⁻/CD45⁻ cells on day 14 of RBC differentiation as determined by flow cytometry. Points are shown as median ± interquartile range. *n* = 3 biologically independent HSPC donors. **c**, Percentage of edited alleles at *HBA1* in all multiplexed editing or spike-in conditions throughout RBC differentiation. Points are shown as median ± 95% confidence interval. *n* = 3 biologically independent HSPC donors. *\*P* = 0.0332 for *HBB* + *tEPOR* 100%, *\*P* = 0.0086 for *HBB* + *tEPOR* 30%, *\*P* = 0.0122 for *HBB* + *tEPOR*

10% (day 0 versus day 14) by unpaired two-tailed *t*-test; *\*\*P* = 0.0113 for *HBB* versus *HBB* + *tEPOR* 30% at day 14, *\*\*P* = 0.008 for *HBB* versus *HBB* + *tEPOR* 10% at day 14 by unpaired two-tailed *t*-test. **d**, Fold increase in edited alleles on day 14 of differentiation of multiplexed conditions versus *HBB* only. The dashed line represents no fold change. Bars represent median ± 95% confidence interval. *\*P* = 0.0315, *\*\*P* = 0.0123 by unpaired two-tailed *t*-test. **e**, mRNA expression of integrated *HBB* at *HBA1* locus normalized to *HBB* expression from mock. *GPA* mRNA expression was used as a reference. *n* = 3 biologically independent HSPC donors. Bars represent median ± 95% confidence interval.

conditions (Fig. 4b). In all the multiplexed conditions, we observed an increase in the frequency of corrected alleles over the course of RBC differentiation (*P* = 0.0332 for *HBB* + *tEPOR* 100%, *P* = 0.0086 for *HBB* + *tEPOR* 30% and *P* = 0.0122 for *HBB* + *tEPOR* 10% when comparing day 0 with day 14; Fig. 4c). In fact, in both the 30% and 10% multiplexed conditions, we achieved a higher frequency of edited alleles by the conclusion of erythroid differentiation compared with single-edited conditions with unedited cells introduced at an equivalent concentration (*P* = 0.0113 for *HBB* versus *HBB* + *tEPOR* 30% at day

14 and *P* = 0.008 for *HBB* versus *HBB* + *tEPOR* 10% at day 14; Fig. 4d). We confirmed truncation at the *EPOR* locus measured by GFP⁺ cells at day 14 of RBC differentiation (Extended Data Fig. 8a). We also observed a corresponding increase in *HBB* mRNA expression from the *HBA1* locus in the 30% and 10% multiplexed edited conditions on day 14 of RBC differentiation compared with single-edited conditions (Fig. 4e), indicating an improved ability for multiplexed editing to increase therapeutic potential of this β-thalassaemia correction strategy. When we measured the colony-forming ability of the edited cells, we found

that differentiation into the various lineages was similar between mock cells and cells edited with *HBB* alone or *HBB* + *tEPOR*; however, we did observe a decrease in total colonies produced in both edited conditions as would be expected from the increased amount of AAV used to target two loci (Extended Data Fig. 8b,c)[34,35].

## Discussion

Insights from clinical genetics have typically implicated new genes and pathways in disease. Here we sought to use human genetics to develop new strategies to treat disease. We used the precision of genome editing to capitalize on a previously characterized disorder called CE, which leads to EPO hypersensitivity and hyper-production of erythrocytes, without causing pathology[4]. As previously shown with variants identified in human genetics, such as those found in *CCR5*, *PCSK9* and the *γ-globin* promoter region, we hypothesized that we could use CRISPR to introduce this natural *EPOR* variant (*tEPOR*) to increase erythropoietic output from edited HSPCs. Previous work has highlighted the challenge of achieving long-term correction of disease following delivery of gene therapy or genome-editing correction strategies[2,36,37]. Although many efforts are underway to improve editing and engraftment frequencies, we hypothesized that we could develop a strategy to increase production of the clinically relevant cell type—the RBC—from edited HSPCs. If successful, then lower editing and engraftment frequencies could yield sufficient production of RBCs to achieve therapeutic benefit and thus be curative for patients. Previous work has shown that introducing the *tEPOR* variant using lentiviral delivery enhanced the efficacy of small hairpin RNA knockdown of *BCL11A* in upregulating HgbF, confirming the beneficial function of this variant[9]. However, our work deploys the precision of genome editing to generate tEPOR, which may have broad utility across a spectrum of blood disorders.

We explored multiple genome-editing strategies to create the *tEPOR* variant, either through truncation of the endogenous *EPOR* gene or integration of a *tEPOR* cDNA at safe-harbour loci. We found that HSPCs expressing *tEPOR* consistently showed increased erythropoiesis but otherwise normal, EPO-dependent production of haemoglobin. To increase RBC production of genome-edited HSPCs in the context of disease correction, we combined the *tEPOR* cassette with a previously described β-thalassaemia strategy in which the *HBB* gene replaces the *HBA1* gene using HDR-based genome editing[14]. This allowed us to simultaneously introduce an *HBB* transgene to restore normal haemoglobin production and to increase erythropoietic output from edited HSPCs. To show the flexibility of the various *tEPOR*-introduction strategies, we also developed an alternative, multiplexed, site-specific genome-editing strategy to pair the original β-thalassaemia correction strategy with introduction of the *EPOR* truncation at the endogenous locus. Both strategies led to enrichment of genome-edited RBCs over the course of erythroid differentiation compared with the traditional β-thalassaemia correction strategy. As we found the effects to be EPO dependent, it is possible the selective advantage in vivo may be even more pronounced because patients suffering from the haemoglobinopathies display elevated EPO levels due to their severe anaemia[38].

In terms of safety, the strategy to use CRISPR to introduce natural variants has the benefit of having been already 'tested' in vivo in humans. However, it must be noted that several of the genome-editing strategies introduce *tEPOR* under non-native regulation that could alter the normal function of tEPOR. In considering this possibility, we note that although every cell in patients with CE harbours an *EPOR* truncation, therapeutic deployment of the genome-editing strategy will result in the introduction of *tEPOR* in only a subset of HSPCs resident in the BM. Therefore, any aberrant effects of non-native *tEPOR* expression (such as bias away from lymphoid or other cell types) are unlikely to lead to cytopenia given the large number of unedited HSPCs remaining in the BM post-transplant. Furthermore, by increasing erythropoietic output

from edited HSPCs, we believe this work could enable the reduction or elimination of high-morbidity myeloablation regimens that are currently required to attain therapeutic levels of edited HSPCs. Expression of *tEPOR* could therefore be integrated into any treatment for blood disorders that involve transplantation of HSPCs. For example, even in an allogeneic haematopoietic stem cell transplant for RBC disorders, a truncation in the natural *EPOR* could be created using indel-based genome editing to give the derived transplanted erythroid progenitors a selective advantage. This strategy could thereby enable less toxic myeloablative conditioning to be effectively used where mixed chimerism might be the result.

Taken together, these results show the power of combining knowledge from human genetics with the precision of CRISPR genome-editing technology to introduce clinically meaningful variants. As human genome sequencing becomes more commonplace and clinically routine[39,40], it is likely that a greater number of variants of unknown significance will be discovered and characterized. We therefore believe that the strategy defined in this work—using CRISPR to introduce natural human variants—may be deployed to amplify the therapeutic potential of current and future cell therapies.

## Methods

### AAV6 vector design, production and purification

AAV6 vector plasmids were cloned into the pAAV-MCS plasmid (Agilent Technologies) comprising inverted terminal repeats derived from AAV2. Gibson Assembly Master Mix (New England Biolabs) was used for the creation of all DNA repair vectors as per the manufacturer's instructions. AAV6 vector was produced and purified with little variation from previously described processes[41]. 293T cells (Life Technologies) were seeded in five dishes (15 cm$^2$) with $13 \times 10^6$–$15 \times 10^6$ cells per plate at 24-h pretransfection. Each dish was then transfected with a standard polyethylenimine (PEI) transfection of 6 μg inverted-terminal-repeat-containing plasmid and 22 μg pDGM6 (gift from David Russell, University of Washington), which holds the AAV6 *cap*, AAV2 *rep* and Ad5 *helper* genes. After a 48–72-h incubation, cells were collected and vectors were purified using the AAVpro purification kit (catalogue number 6666; Takara Bio) per the manufacturer's instructions and then stored at −80 °C until further use. AAV6 vectors were titred using ddPCR to measure the number of vector genomes as previously described[42].

### In vitro culture of CD34$^+$ HSPCs

Human CD34$^+$ HSPCs were cultured in conditions as previously described[13,43–46]. CD34$^+$ HSPCs were isolated from cord blood (provided by Stanford Binns Program for Cord Blood Research) or sourced from plerixafor- and/or G-CSF-mobilized peripheral blood (AllCells and STEMCELL Technologies). Frozen plerixafor- and/or G-CSF-mobilized peripheral blood of patients with SCD were provided by Dr Vivien Sheehan at Emory University. CD34$^+$ HSPCs were cultured at $1 \times 10^5$–$5 \times 10^5$ cells ml$^{-1}$ in StemSpan Serum-Free Expansion Medium II (STEMCELL Technologies) or Good Manufacturing Practice Stem Cell Growth Medium (SCGM; CellGenix) supplemented with a human cytokine (PeproTech) cocktail: stem cell factor (100 ng ml$^{-1}$), thrombopoietin (100 ng ml$^{-1}$), Fms-like tyrosine kinase 3 ligand (100 ng ml$^{-1}$), interleukin-6 (100 ng ml$^{-1}$), streptomycin (20 mg ml$^{-1}$), penicillin (20 U ml$^{-1}$) and 35 nM of UM171 (catalogue number A89505; APExBIO). The cell incubator conditions were 37 °C, 5% CO$_2$ and 5% O$_2$.

### Electroporation-aided transduction of cells

The synthetic chemically modified sgRNAs used to edit CD34$^+$ HSPCs were purchased from Synthego or TriLink Biotechnologies and were purified by HPLC. These modifications comprise 2′-*O*-methyl-3′-phosphorothioate at the three terminal nucleotides of the 5′ and

3′ ends described previously[17]. The target sequences for the gRNAs were as follows.

**EPOR gRNA (EPOR-sg1).**
5′-AGCTCAGGGCACAGTGTCCA-3′

**EPOR gRNA (EPOR-sg2).**
5′-GCTCCCAGCTCTTGCGTCCA-3′

**CCR5 gRNA (CCR5-sg3).**
5′-GCAGCATAGTGAGCCCAGAA-3′

**HBA1 gRNA (HBA1-sg4).**
5′-GGCAAGAAGCATGGCCACCG-3′

The HiFi Cas9 protein was purchased from Integrated DNA Technologies (IDT) or Aldevron. Before electroporation, RNPs were complexed at a Cas9/sgRNA molar ratio of 1:2.5 at 25 °C for 10–20 min. Next, CD34+ cells were resuspended in P3 buffer (Lonza) with complexed RNPs and subsequently electroporated using the Lonza 4D-Nucleofector and 4D-Nucleofector X Unit (program DZ-100). Electroporated cells were then plated at $1 \times 10^5$–$5 \times 10^5$ cells ml$^{-1}$ in the previously described cytokine-supplemented media. Immediately after electroporation, AAV6 was dispensed onto cells at $2.5 \times 10^3$–$5 \times 10^3$ vector genomes per cell based on titre determined by ddPCR. For multiplex editing experiments, in addition to the steps described above, cells were incubated with 0.5 μM of the DNA-PKcs inhibitor AZD7648 (catalogue number S8843; Selleck Chemicals) for 24 h, as previously described[32,33].

### Allelic modification analysis using ddPCR
Edited HSPCs were collected within 2–3 days postelectroporation and at each media change throughout erythrocyte differentiation and then analysed for modification frequencies of the alleles of interest. To quantify editing frequencies, we created custom ddPCR primers and probes to quantify HDR alleles (using in–out PCR and probe corresponding to the expected integration event) compared with an established genomic DNA reference (REF) at the CCRL2 locus[14]. QuickExtract DNA extraction solution (catalogue number QE09050; Biosearch Technologies) was used to collect genomic DNA input, which was then digested using BamHI-HF or HindIII-HF as per the manufacturer's instructions (New England Biolabs). The percentage of targeted alleles within a cell population was measured with a Bio-Rad QX200 ddPCR machine and QuantaSoft software (v.1.7; Bio-Rad) using the following reaction mixture: 1–4 μl genomic DNA input, 10 μl of ddPCR Supermix for Probes (no dUTP; Bio-Rad), primer and probes (1:3.6 ratio; IDT), and volume up to 20 μl with $H_2O$. ddPCR droplets were then generated following the manufacturer's instructions (Bio-Rad): 20 μl of ddPCR reaction, 70 μl of droplet generation oil and 40 μl of droplet sample. Thermocycler (Bio-Rad) settings were as follows: 98 °C (10 min), 94 °C (30 s), 55.7–60 °C (30 s), 72 °C (2 min), return to step 2 for 40–50 cycles and then 98 °C (10 min). Analysis of droplet samples was then performed using the QX200 Droplet Digital PCR System (Bio-Rad). We next divided the copies per microlitre for HDR (%): HDR/REF. The following primers and probes were used in the ddPCR reaction.

**CCR5 (for tEPOR-YFP construct).**
Forward primer (FP): 5′-GGGAGGATTGGGAAGACA-3′
Reverse primer (RP): 5′-AGGTGTTCAGGAGAAGGACA-3′
Probe: 5′-6-FAM/AGCAGGCATGCTGGGGATGCGGTGG/3IABkFQ-3′

**HBA1 (for tEPOR-YFP construct).**
FP: 5′-AGTCCAAGCTGAGCAAAGA-3′
RP: 5′-ATCACAAACGCAGGCAGAG-3′
Probe: 5′-6-FAM/CGAGAAGCGCGATCACATGGTCCTGC/3IABkFQ-3′

**HBA1 (for HBB construct and tEPOR-HBB constructs).**
FP: 5′-GTGGCTGGTGTGGCTAATG-3′
RP: 5′-CAGAAAGCCAGCCAGTTCTT-3′
Probe: 5′-6-FAM/CCTGGCCCACAAGTATCACT/3IABkFQ-3′

**HBA1 (for HBB-tEPOR constructs).**
FP: 5′-TCTGCTGCCAGCTTTGAGTA-3′
RP: 5′-GCTGGAGTGGGACTTCTCTG-3′
Probe: 5′-6-FAM/ACTATCCTGGACCCCAGCTC/3IABkFQ-3′

**CCRL2 (reference).**
FP: 5′-GCTGTATGAATCCAGGTCC-3′
RP: 5′-CCTCCTGGCTGAGAAAAAG-3′
Probe: 5′-HEX/TGTTTCCTC/ZEN/CAGGATAAGGCAGCTGT/3IABkFQ-3′

### Indel analysis using TIDE software
Within 2–4 days postelectroporation, HSPCs were collected with QuickExtract DNA extraction solution (catalogue number QE09050; Biosearch Technologies) to collect genomic DNA. The following primer sequences were used to amplify the respective cut sites at the EPOR locus:
FP: 5′-CAGCTGTGGCTGTACCAGAA-3′
RP: 5′-CAGCCTGGTGTCCTAAGAGC-3′
Sanger sequencing of the respective samples was then used as input for indel frequency analysis relative to a mock, unedited sample using TIDE as previously described[25].

### In vitro differentiation of CD34+ HSPCs into erythrocytes
Following editing, HSPCs derived from healthy individuals or patients with SCD were cultured for 2–3 days as described above. Subsequently, a 14-day in vitro differentiation was performed in supplemented SFEMII medium as previously described[24,47]. SFEMII base medium was supplemented with 100 U ml$^{-1}$ penicillin–streptomycin, 10 ng ml$^{-1}$ SCF (PeproTech), 1 ng ml$^{-1}$ IL-3 (PeproTech), 3 U ml$^{-1}$ EPO (eBiosciences), 200 μg ml$^{-1}$ transferrin (Sigma-Aldrich), 3% human serum (heat-inactivated; Sigma-Aldrich or Thermo Fisher Scientific), 2% human plasma (isolated from umbilical cord blood provided by the Stanford Binns Cord Blood Program), 10 μg ml$^{-1}$ insulin (Sigma-Aldrich) and 3 U ml$^{-1}$ heparin (Sigma-Aldrich). Cells were cultured in the first phase of medium for 7 days at $1 \times 10^5$ cells ml$^{-1}$. In the second phase of medium, days 7–10, cells were maintained at $1 \times 10^5$ cells ml$^{-1}$ and IL-3 was removed from the culture. In the third phase of medium, days 11–14, cells were cultured at $1 \times 10^6$ cells ml$^{-1}$, with a transferrin increase to 1 mg ml$^{-1}$.

### Immunophenotyping of differentiated erythrocytes
Differentiated erythrocytes were analysed by flow cytometry on day 14 for erythrocyte lineage-specific markers using a FACS Aria II (BD Biosciences). Edited and unedited cells were analysed using the following antibodies: hCD45-V450 (HI30; BD Biosciences), CD34-APC (561; BioLegend), CD71-PE-Cy7 (OKT9; Affymetrix) and CD235a-PE (GPA) (GA-R2; BD Biosciences). In addition to cell-specific markers, cells were also stained with Ghost Dye Red 780 (Tonbo Biosciences) to measure viability.

### Haemoglobin tetramer analysis
Frozen pellets of approximately $1 \times 10^6$ in vitro-differentiated erythrocytes were thawed and lysed in 30 μl of RIPA buffer with 1× Halt Protease Inhibitor Cocktail (Thermo Fisher Scientific) for 5 min on ice. The mixture was vigorously vortexed and cell debris was removed by centrifugation at 13,000 r.p.m. for 10 min at 4 °C. HPLC analysis of haemoglobins in their native form was performed on a cation-exchange PolyCAT A column (35 mm² × 4.6 mm², 3 μm, 1,500 Å; PolyLC) using a Perkin-Elmer Flexar HPLC system at room temperature and detection at 415 nm. Mobile phase A consisted of 20 mM Bis-Tris and 2 mM KCN at

pH 6.94, adjusted with HCl. Mobile phase B consisted of 20 mM Bis-Tris, 2 mM KCN and 200 mM NaCl at pH 6.55. Haemolysate was diluted in buffer A before injection of 20 µl onto the column with 8% buffer B and eluted at a flow rate of 2 ml min$^{-1}$ with a gradient made to 40% B in 6 min, increased to 100% B in 1.5 min, returned to 8% B in 1 min and equilibrated for 3.5 min. Quantification of the area under the curve of the peaks was performed with TotalChrom software (Perkin-Elmer) and raw values were exported to GraphPad Prism 9 for plotting and further analysis.

### mRNA analysis

After differentiation of HSPCs into erythrocytes, cells were collected and RNA was extracted using the RNeasy Plus Mini Kit (Qiagen). Subsequently, cDNA was made from approximately 100 ng of RNA using the iScript Reverse Transcription Supermix for quantitative PCR with reverse transcription (Bio-Rad). Expression levels of the *β-globin* transgene and *α-globin* mRNA were quantified with a Bio-Rad QX200 ddPCR machine and QuantaSoft software (v.1.7; Bio-Rad) using the following primers and 6-FAM/ZEN/IBFQ-labelled hydrolysis probes, purchased as custom-designed PrimeTime qPCR Assays from IDT.

#### *HBB* and *HBB-tEPOR* into *HBA1*.
FP: 5′-GGTCCCCACAGACTCAGAGA-3′
RP: 5′-CAGCATCAGGAGTGGACAGA-3′
Probe: 5′-6-FAM/AACCCACCATGGTGCATCTG/3IABkFQ-3′

To normalize for RNA input, levels of the RBC-specific reference gene *GPA* were determined in each sample using the following primers and HEX/ZEN/IBFQ-labelled hydrolysis probes, purchased as custom-designed PrimeTime qPCR Assays from IDT.

#### *GPA* (reference).
FP: 5′-ATATGCAGCCACTCCTAGAGCTC-3′
RP: 5′-CTGGTTCAGAGAAATGATGGGCA-3′
Probe: 5′-HEX/AGGAAACCGGAGAAAGGGTA/3IABkFQ-3′

ddPCR reactions were created using the respective primers and probes and droplets were generated as described above. Thermocycler (Bio-Rad) settings were as follows: 98 °C (10 min), 94 °C (30 s), 54 °C (30 s), 72 °C (30 s), return to step 2 for 50 cycles and then 98 °C (10 min). Analysis of droplet samples was done using the QX200 Droplet Digital PCR System (Bio-Rad). To determine relative expression levels, the numbers of *HBB* transgene copies per millilitre were divided by the numbers of *GPA* copies ml$^{-1}$.

### Methylcellulose CFU assay

At 2–3 days postelectroporation, HSPCs were plated in SmartDish 6 well plates (catalogue number 27370; STEMCELL Technologies) containing MethoCult H4434 Classic or MethoCult H4434 Classic without EPO (catalogue numbers 04444 and 04544; STEMCELL Technologies). After 14 days, the wells were imaged using the STEMvision Hematopoietic Colony Counter (STEMCELL Technologies). Colonies were counted and scored to determine the number of CFU-GEMM (colony-forming unit-granulocyte, erythroid, macrophage, megakaryocyte), CFU-GM (colony-forming unit-granulocyte, macrophage), BFU-E (burst-forming unit-erythroid) and CFU-E (colony-forming unit-erythroid) colonies.

### Quantification of editing efficiency at evaluated off-target sites

Potential sgRNA off-target sites were predicted using the CRISPR Off-target Sites with Mismatches, Insertions and Deletions (COSMID) online tool[48]. Sites were ranked according to score and duplicate predictions at the same location were removed. All sites with a score ≤5.5 were included in the analysis and the 5 sites in exonic or untranslated regions were further analysed. PCR amplification of these sites was performed using genomic DNA from mock-edited and RNP-edited cells. The following primers were used with Illumina adaptors (FP adaptor, 5′-ACACTCTTTCCCTACACGACGCTCTTCCGATCT-3′; RP adaptor, 5′-GACTGGAGTTCAGACGTGTGCTCTTCCGATCT-3′).

#### *EPOR-OT1.*
FP: 5′-GAGCGGGCTACAGAGCTAGA-3′
RP: 5′-TGGCAGAAAGTAAGGGGATG-3′

#### *EPOR-OT2.*
FP: 5′-ACTTGTGGAGCCACAGTTTG-3′
RP: 5′-AATGCCCTTGAGATGAATGC-3′

#### *EPOR-OT3.*
FP: 5′-TCACACACCCGTAGCCATAA-3′
RP: 5′-AGAATGCTCTTTGCAGTAGCC-3′

#### *EPOR-OT4.*
FP: 5′-CTCAAAACTTCACCCAGGCT-3′
RP: 5′-GGTCTGTCATTGAATGCCTT-3′

#### *EPOR-OT5.*
FP: 5′-CAACCCTGATGGGTCTGC-3′
RP: 5′-CCACAGCTGGCTGACCTT-3′

Following amplification, PCR products were purified by gel electrophoresis and subsequent extraction using the GeneJet Gel Extraction Kit (catalogue number FERK0692; Thermo Fisher Scientific). Purified samples were submitted for library preparation and sequencing by Amplicon-EZ next-generation sequencing (Azenta Life Sciences), ensuring a yield of over 100,000 reads per sample. Amplicons, flanked by Illumina partial adaptor sequences, which encompassed the programmed DSBs for CRISPR–Cas9, underwent sequencing using Illumina chemistry. FastQC (v.0.11.8, default parameters; http://www.bioinformatics.babraham.ac.uk/projects/fastqc/) was used to assess the quality of raw reads. Subsequently, paired-end reads were aligned to the specified off-target regions using CRISPResso2 (v.2.2.14; fastq.gz files were used as input)[49].

### Statistical analysis

GraphPad Prism 9 software was used for all statistical analysis.

### Reporting summary

Further information on research design is available in the Nature Portfolio Reporting Summary linked to this article.

## Data availability

The main data supporting the results in this study are available within the article and its Supplementary Information. High-throughput-sequencing data generated for off-target analysis are available through the NCBI Sequence Read Archive database via the accession number PRJNA1102034. Sequences of the gRNA and ddPCR primers and probes are provided in Methods. Source data for the figures are provided with this paper. All data generated in this study are available from the corresponding authors on reasonable request.

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

## Acknowledgements

We thank the following funding sources that made this work possible: American Society of Hematology Junior Faculty Scholar Award to M.K.C.; Stanford Medical Scholars Research Program, the American Society of Hematology Minority Medical Student Award Program and the Stanford Medical Scientist Training Program to S.E.L.; and American Society of Hematology Minority Medical Student Award Program to J.C. We thank the Stanford Binns Program for Cord Blood Research for providing CD34⁺ HSPCs and the FACS Core Facility at the Stanford Institute for Stem Cell Biology and Regenerative Medicine for access to FACS machines.

## Author contributions

S.E.L., J.C., J.P.H., K.R.M., C.T.C., E.S. and M.K.C. designed and performed experiments and analysed data. S.S. and K.J. designed experiments and analysed data. V.A.S. provided reagents for experiments. M.H.P. and M.K.C. supervised the project. S.E.L., J.C. and M.K.C. wrote the article with input from other authors.

## Competing interests

M.H.P. serves on the scientific advisory board of Allogene Tx and is an advisor to Versant Ventures. M.H.P., M.K.C. and J.C. have equity in Graphite Bio. M.H.P. has equity in CRISPR Tx and serves on the board of directors of Graphite Bio and Kamau Therapeutics. J.C., M.K.C. and M.H.P. have filed provisional patent number PCT/US2022/077951. All other authors declare no competing interests.

## Additional information

**Extended data** is available for this paper at https://doi.org/10.1038/s41551-024-01222-6.

**Correspondence and requests for materials** should be addressed to M. Kyle Cromer or Matthew H. Porteus.

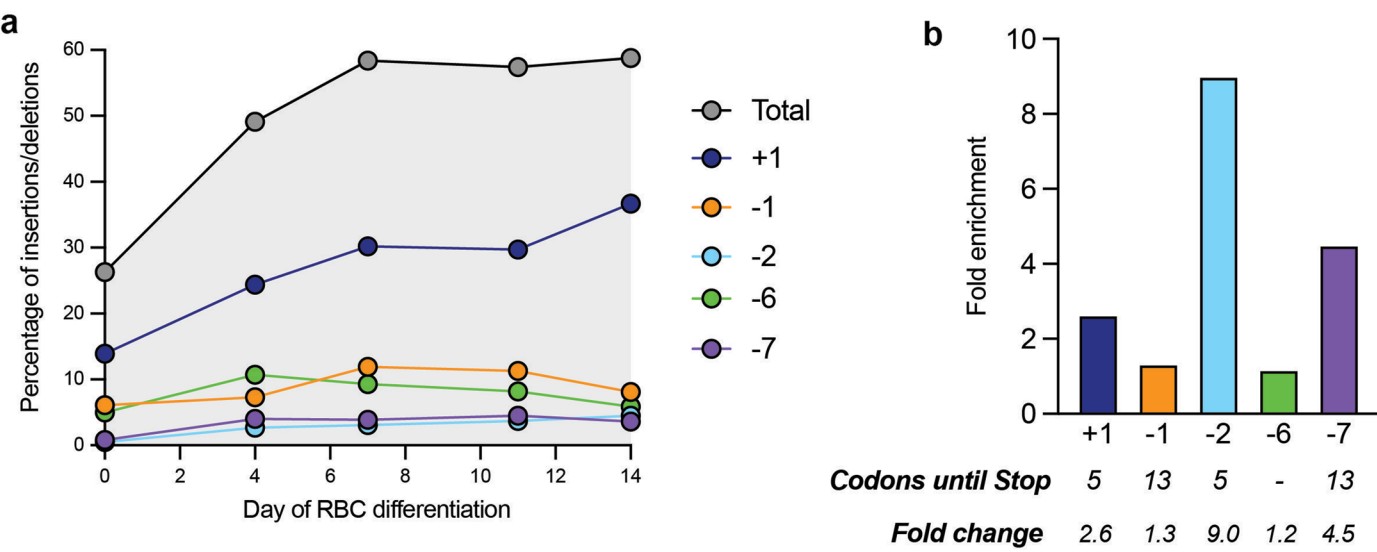

**Extended Data Fig. 1 | Indel analysis of HSPCs edited with most effective *EPOR* sgRNA (sg1) over course of erythroid differentiation. a**, Frequency of five most common indels found in one HSPC donor targeted with *EPOR*-sg1 over the course of RBC differentiation. **b**, Fold enrichment of five most common indels over course of RBC differentiation in one HSPC donor targeted with *EPOR*-sg1.

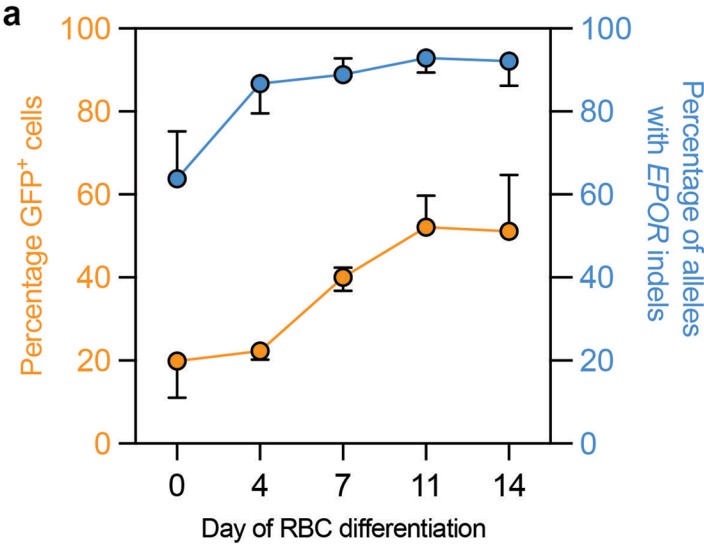

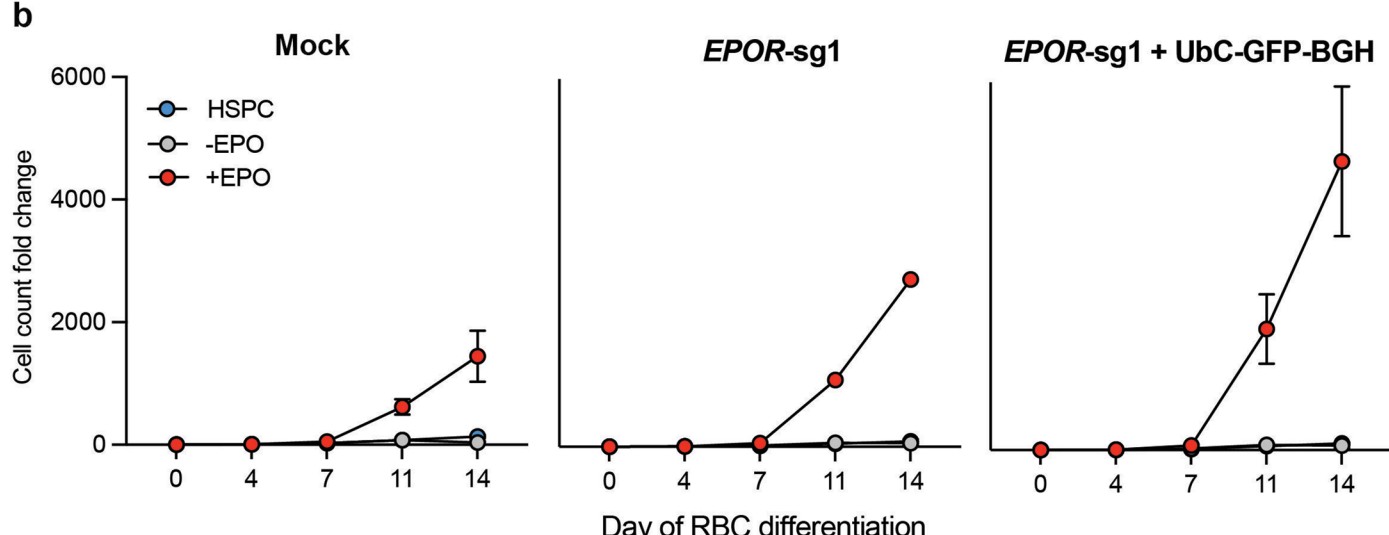

**Extended Data Fig. 2 | Almost all cells edited at *EPOR* locus contain tEPOR at end of differentiation and show increased erythroid proliferation. a**, Plot of percentage of GFP+ cells of live single cells maintained in RBC media +EPO (from data shown in Fig. 1e) overlaid with percentage of alleles containing indels in *EPOR* in same three biological donors. Points represent median ± 95% confidence interval. **b**, Cell count fold change in mock and edited cells maintained in RBC media +/- EPO or HSPC media. Points represent mean ± SEM. Values represent biologically independent HSPC donors N = 3 for Mock and *EPOR*-sg1+ BGH and N = 2 for sg1. Data for +EPO condition same as in Fig. 1f.

**a**

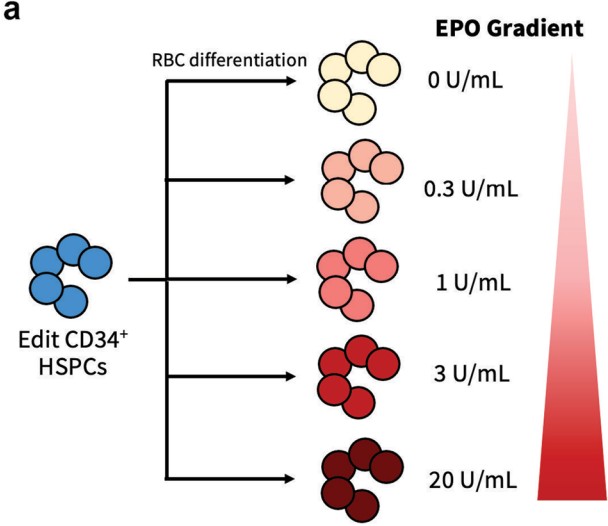

**b**

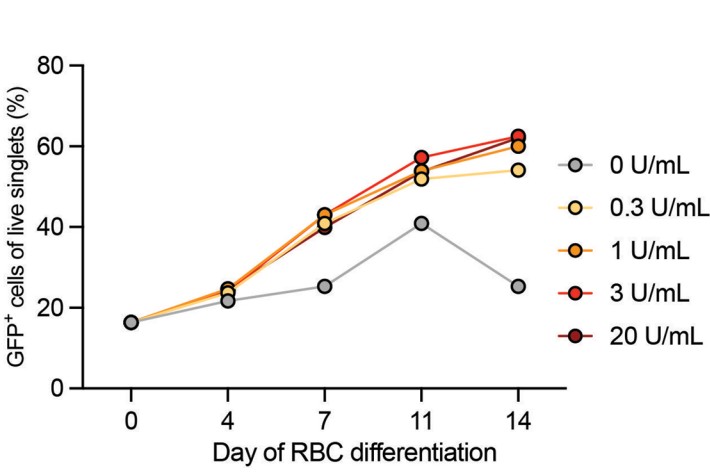

**c**

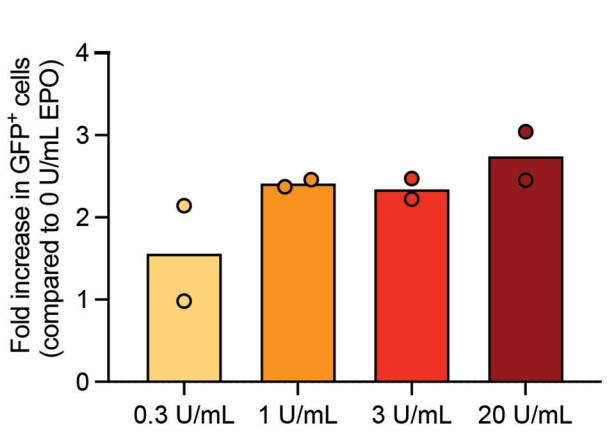

**d**

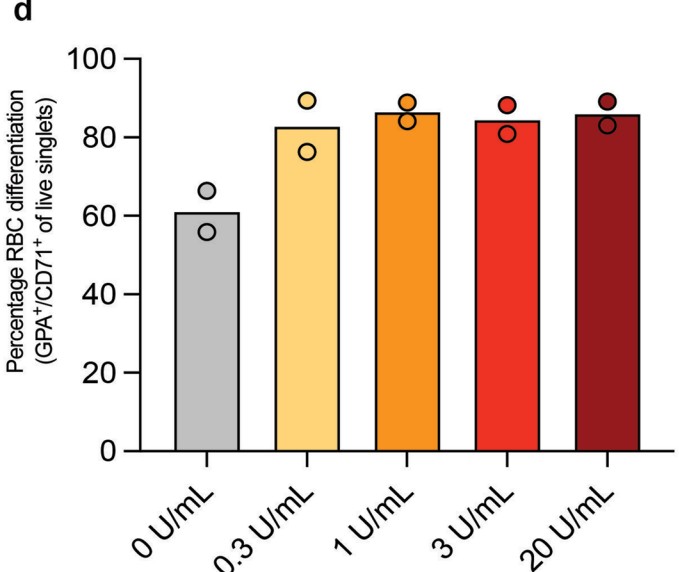

**Extended Data Fig. 3 | Enrichment of edited cells and RBC differentiation is only minorly affected by different concentration of EPO in media.**
**a**, Schematic depicting HSPC editing and subsequent RBC differentiation using different levels of EPO cytokine. **b**, Percentage of GFP+ cells of live single cells maintained in RBC media with 0 U/mL – 20 U/mL EPO from one biological HSPC donor. **c**, Fold increase of GFP+ cells at each concentration of EPO compared to 0 U/mL EPO at day 14 of RBC differentiation. Bars represent median ± 95% confidence interval. N = 2 biological HSPC donors. **d**, Percentage of GPA+/CD71+ of live single cells on day 14 of differentiation. Bars represent median ± 95% confidence interval. Values represent N = 2 biologically independent HSPC donors.

**a** Predicted sites with score ≤ 5.5

| | Chromosomal Position | Score | Region | Gene |
|---|---|---|---|---|
| On-target | Chr19:11378177-11378199 | 0 | Exon | *EPOR* |
| Off-target | Chr4:2834819-2834841 | 0.54 | 5' UTR | *SH3BP2* |
| | Chr17:63145508-63145530 | 0.61 | Intergenic | |
| | Chr14:74552967-74552989 | 0.63 | Exon | *LTBP2* |
| | Chr17:76190490-76190511 | 1.02 | Intron | *RNF157* |
| | Chr17:74795165-74795187 | 1.08 | Intron | *TMEM104* |
| | Chr19:57614583-57614605 | 1.22 | Intron | *ZNF134* |
| | Chr22:23681747-23681770 | 1.25 | Intron | *GUSBP11* |
| | Chr9:133742339-133742361 | 1.44 | Intergenic | |
| | Chr1:109777058-109777079 | 1.48 | Intergenic | |
| | Chr17:3725613-3725635 | 1.5 | Exon | *HASPIN* |
| | Chr20:47957087-47957110 | 1.51 | Intergenic | |
| | Chr15:62897167-62897188 | 1.52 | Intergenic | |
| | Chr1:153353786-153353807 | 1.53 | Intergenic | |
| | Chr22:47588374-47588397 | 1.79 | Intergenic | |
| | Chr5:66475949-66475972 | 1.82 | Intergenic | |
| | Chr15:35375033-35375054 | 1.83 | Intron | *DPH6* |
| | Chr3:146694160-146694183 | 2.08 | Intergenic | |
| | ChrX:53455142-53455163 | 2.15 | Intergenic | |
| | Chr16:58089238-58089259 | 2.15 | Intergenic | |
| | Chr15:66336597-66336618 | 2.15 | 3' UTR | *TIPIN* |
| | Chr11:69809008-69809031 | 2.17 | Intergenic | |
| | Chr8:125489832-125489853 | 2.21 | Intergenic | |
| | Chr8:1741386-1741407 | 2.35 | Intergenic | |
| | Chr6:74015621-74015642 | 2.35 | Intergenic | |
| | Chr2:207213231-207213252 | 2.66 | Intron | *SLID-AS1* |
| | Chr7:68423063-68423084 | 2.69 | Intergenic | |
| | Chr11:73258289-73258311 | 2.72 | Intergenic | |
| | Chr2:45355014-45355035 | 2.95 | Intergenic | |
| | Chr3:60743768-60743789 | 2.95 | Intron | *FHIT* |
| | Chr6:103967054-103967075 | 3.04 | Intergenic | |
| | Chr3:22886579-22886600 | 3.06 | Intergenic | |
| | Chr16:88736230-88736251 | 3.11 | Exon | *PIEZO1* |
| | Chr2:120715491-120715512 | 3.11 | Intergenic | |
| | ChrX:68993580-68993602 | 3.17 | Intergenic | |
| | Chr5:43348327-43348348 | 3.18 | Intergenic | |
| | Chr9:123359014-123359035 | 3.2 | Intron | *CRB2* |
| | ChrX:30077883-30077904 | 3.21 | Intergenic | |
| | Chr22:34432694-34432715 | 3.21 | Intergenic | |
| | Chr21:45947860-45947882 | 3.31 | Intergenic | |
| | Chr12:94243003-94243025 | 3.35 | Intron | *PLXNC1* |
| | Chr18:79252201-79252223 | 3.4 | Intron | *ATP9B* |
| | Chr8:144201447-144201469 | 3.42 | Intron | *MROH1* |
| | Chr2:3859090-3859111 | 3.48 | Intergenic | |
| | Chr5:23614195-23614216 | 3.56 | Intergenic | |
| | Chr10:44054693-44054715 | 3.62 | Intergenic | |
| | Chr3:59498371-59498393 | 3.62 | Intergenic | |
| | Chr22:48885862-48885883 | 3.63 | Intron | *LINC01310* |
| | Chr2:236161588-236161609 | 3.63 | Intergenic | |
| | Chr13:76821039-76821060 | 3.66 | Intergenic | |
| | ChrX:69384155-69384176 | 3.78 | Intergenic | |
| | Chr6:22043322-22043343 | 3.89 | Intron | *CACS15* |
| | Chr19:54221554-54221575 | 3.89 | Intron | *LILRB3* |
| | Chr19:54241302-54241323 | 3.89 | Intron | *LILRA6* |
| | Chr3:171095217-171095238 | 3.95 | Intron | *TNIK* |
| | Chr17:29069628-29069649 | 3.96 | Intergenic | |
| | Chr5:79849804-79849827 | 4.02 | Intergenic | |
| | Chr13:23653867-23653888 | 4.03 | Intron | *TNFRSF19* |
| | Chr6:31225407-31225428 | 4.03 | Intergenic | |
| | Chr17:3169786-3169807 | 4.09 | Intron | *LOC100288728* |
| | Chr1:209455360-209455383 | 4.12 | Intergenic | |
| | Chr22:42906036-42906057 | 4.2 | Intron | *PACSIN2* |
| | Chr10:48290891-48290913 | 4.36 | Intergenic | |
| | Chr9:136644277-136644299 | 4.41 | Intergenic | |
| | Chr19:57018808-57018830 | 4.5 | Intergenic | |
| | Chr2:25314961-25314982 | 4.66 | Intergenic | |
| | Chr20:39849896-39849917 | 4.8 | Intergenic | |
| | Chr9:1374116-1374137 | 4.82 | Intergenic | |
| | Chr1:223229923-223229945 | 4.88 | Intron | *SUSD4* |
| | Chr6:54602796-54602819 | 4.95 | Intergenic | |
| | Chr14:103074393-103074414 | 4.95 | Intergenic | |
| | Chr16:50151538-50151561 | 4.99 | Intergenic | |
| | Chr7:91541296-91541318 | 5 | Intergenic | |
| | Chr6:52211164-52211187 | 5.08 | Intergenic | |
| | Chr5:122639266-122639287 | 5.14 | intron | *LINC02201* |
| | Chr14:105666438-105666459 | 5.18 | Intergenic | |
| | Chr13:44678373-44678396 | 5.39 | intron | *LINC00407* |
| | Chr1:154810236-154810258 | 5.4 | intron | *KCNN3* |
| | Chr1:47975646-47975668 | 5.49 | Intergenic | |

**b** OT distribution by region in genome

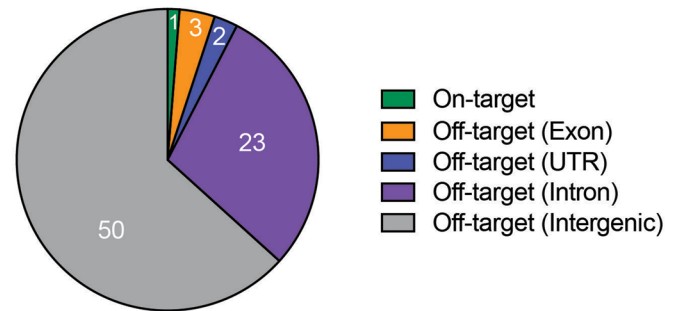

- On-target (1)
- Off-target (Exon) (3)
- Off-target (UTR) (2)
- Off-target (Intron) (23)
- Off-target (Intergenic) (50)

**c** Exonic and UTR OT Sites

| Off-target Site | Chromosomal Position | COSMID Score | Sequence | Mismatch | Strand | Cut Site | Gene | Region of Gene |
|---|---|---|---|---|---|---|---|---|
| OT1 | Chr4:2834819-2834841 | 0.54 | TGCTTACGGCACAGTGTCCAGGG -- hit<br>AGCTCAGGGCACAGTGTCCANGG -- query | 3 | + | 2834835 | *SH3BP2* | 5' UTR |
| OT2 | Chr14:74552967-74552989 | 0.63 | AGCTGTTGGCACAGTGTCCACGG -- hit<br>AGCTCAGGGCACAGTGTCCANGG -- query | 3 | + | 74552983 | *LTBP2* | Exon |
| OT3 | Chr17:3725613-3725635 | 1.5 | AACTCAGTGCACTGTGTCCAGGG -- hit<br>AGCTCAGGGCACAGTGTCCANGG -- query | 3 | + | 3725629 | *HASPIN* | Exon |
| OT4 | Chr15:66336597-66336618 | 2.15 | A^CTCTGGGCACACTGTCCATGG -- hit<br>AGCTCAGGGCACAGTGTCCANGG -- query | 2 | + | 66336612 | *TIPIN* | 3' UTR |
| OT5 | Chr16:88736230-88736251 | 3.11 | AGCTCAGGGCCAG^GTCCATGG -- hit<br>AGCTCAGGGCACAGTGTCCANGG -- query | 1 | + | 88736245 | *PIEZO1* | Exon |

**d**

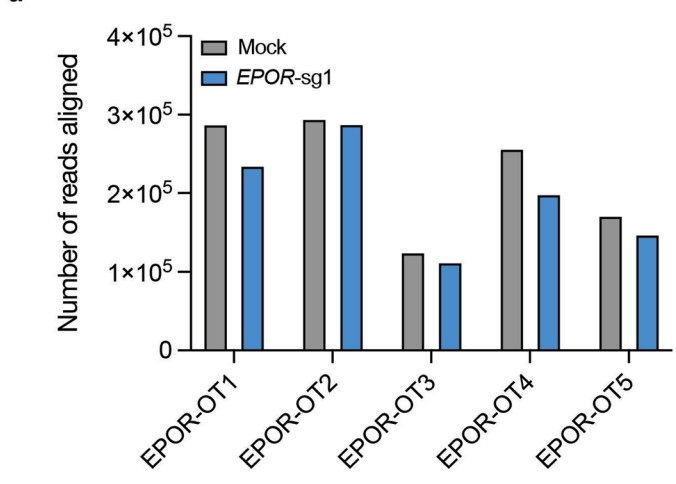

**e**

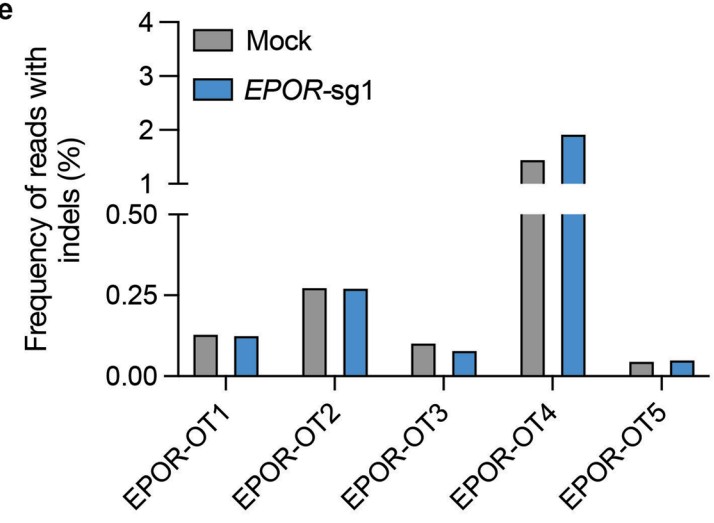

**Extended Data Fig. 4 | See next page for caption.**

**Extended Data Fig. 4 | Off-target analysis of *EPOR*-sg1 by *in silico* prediction and targeted next-generation sequencing (NGS). a**, Complete list of potential off-target sites predicted by COSMID with a score ≤5.5. **b**, Summary of off-target sites classified by region in genome. **c**, Detailed summary of 5 candidate off-target sites found in an exon or UTR of the genome. Sequence nucleotide mismatches highlighted in red, inserted nucleotides underlined, and missing nucleotides indicated by the '^' symbol. PAM sequence highlighted in blue. **d**, Sequencing read depth of mock-edited and *EPOR*-sg1 edited sample sent for each off-target site. N = 1 biological HSPC donor. **e**, Frequency of indels detected in each sample at each off-target site. N = 1 biological HSPC donor.

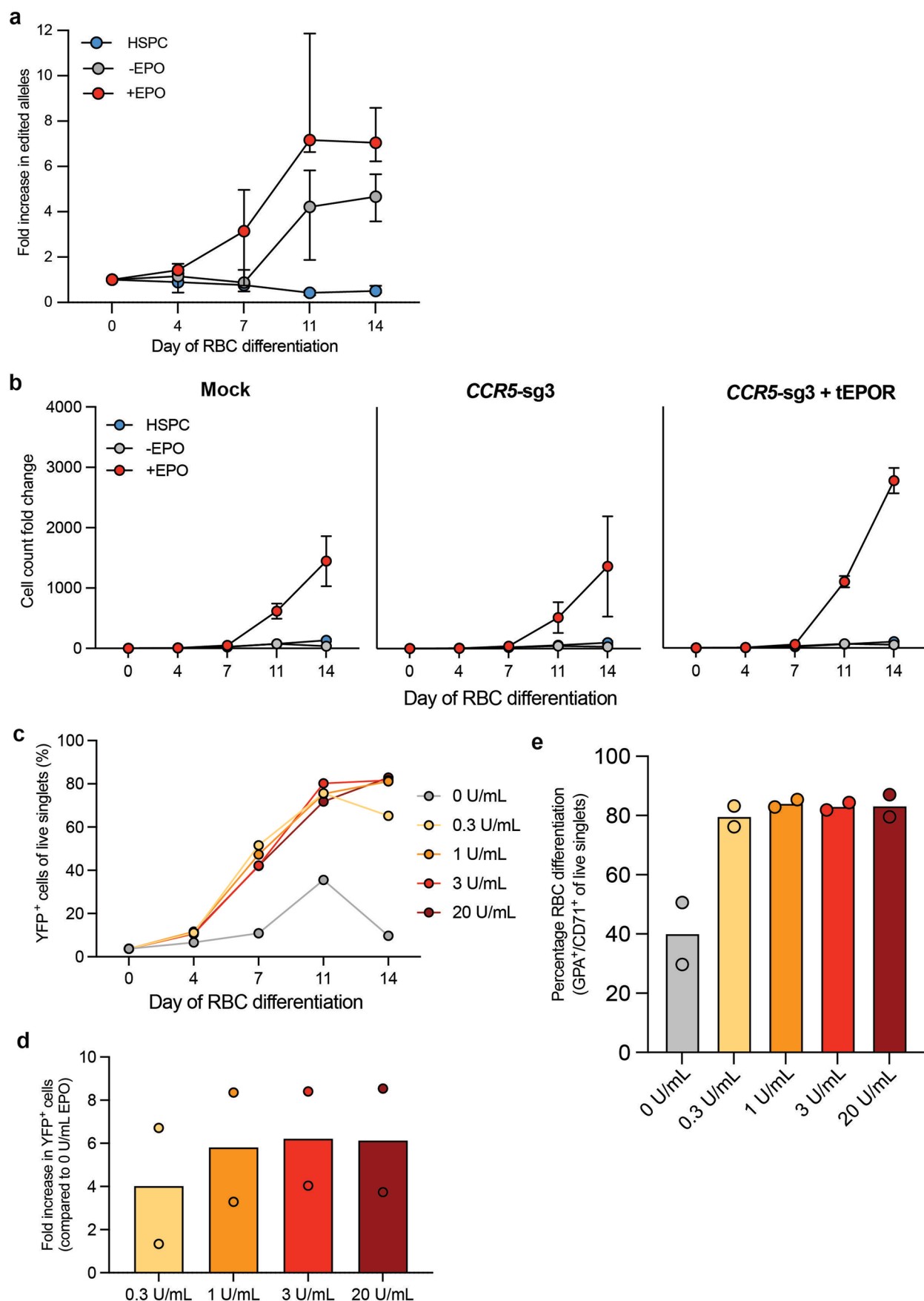

**Extended Data Fig. 5 | See next page for caption.**

**Extended Data Fig. 5 | Safe harbour integration of *tEPOR* leads to increased erythroid proliferation and editing frequencies with little impact at different concentrations of EPO. a**, Fold change in edited alleles at *CCR5* over course of RBC differentiation +/- EPO or maintained in HSPC media measured by ddPCR. Points shown as median ± interquartile range. N = 3-4 biologically independent HSPC donors. **b**, Cell count fold change in mock and edited cells maintained in RBC media +/- EPO or HSPC media. Points represent mean ± SEM. Values represent biologically independent HSPC donors N = 3 for Mock and

*CCR5*-sg3 + *tEPOR* and N = 2 for *CCR5*-sg3. **c**, Percentage of YFP[+] cells of live single cells throughout RBC differentiation with 0 U/mL – 20 U/mL EPO from one biological HSPC donor. **d**, Fold increase of YFP[+] cells at each concentration of EPO compared to 0 U/mL EPO at day 14 of differentiation. Bars represent median ± 95% confidence interval. N = 2 biological HSPC donors. **e**, Percentage of GPA[+]/CD71[+] of live single cells on day 14 of differentiation. Bars represent median ± 95% confidence interval. Values represent N = 2 biologically independent HSPC donors.

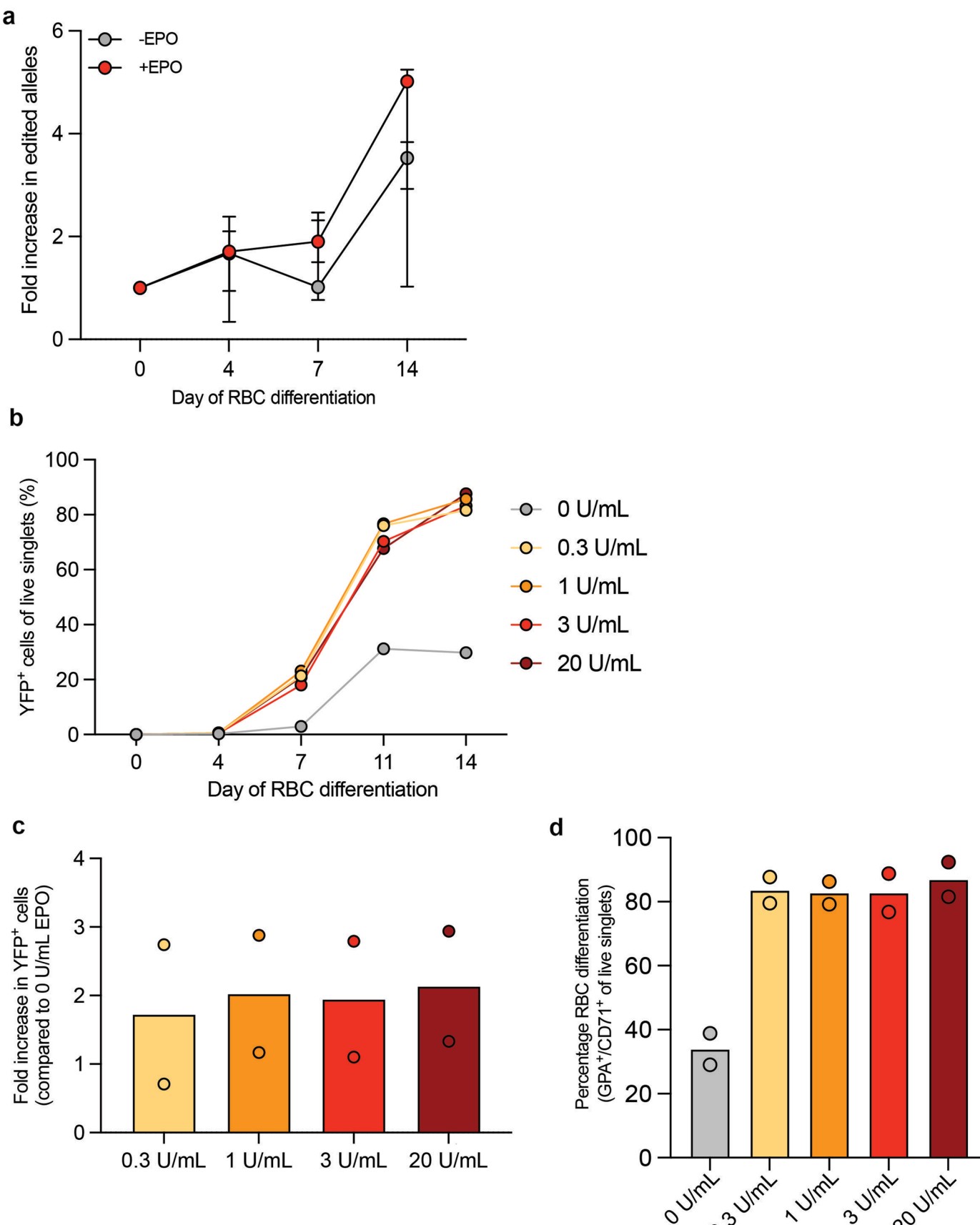

**Extended Data Fig. 6 | See next page for caption.**

**Extended Data Fig. 6 | Erythroid specific expression of *tEPOR* leads to increased editing frequencies and is not affected by EPO concentration.** **a**, Fold change in edited alleles at *HBA1* over course of RBC differentiation +/- EPO measured by ddPCR. Points shown as median ± 95% CI. N = 3 biologically independent HSPC donors. **b**, Percentage of YFP⁺ cells of live single cells throughout RBC differentiation with 0 U/mL – 20 U/mL EPO in one biological HSPC donor. **c**, Fold increase of YFP⁺ cells at each EPO concentration compared to 0 U/mL EPO at day 14 of RBC differentiation. Bars represent median ± 95% confidence interval. N = 2 biological HSPC donors. **d**, Percentage of GPA⁺/CD71⁺ of live single cells on day 14 of differentiation. Bars represent median ± 95% confidence interval. Values represent N = 2 biologically independent HSPC donors.

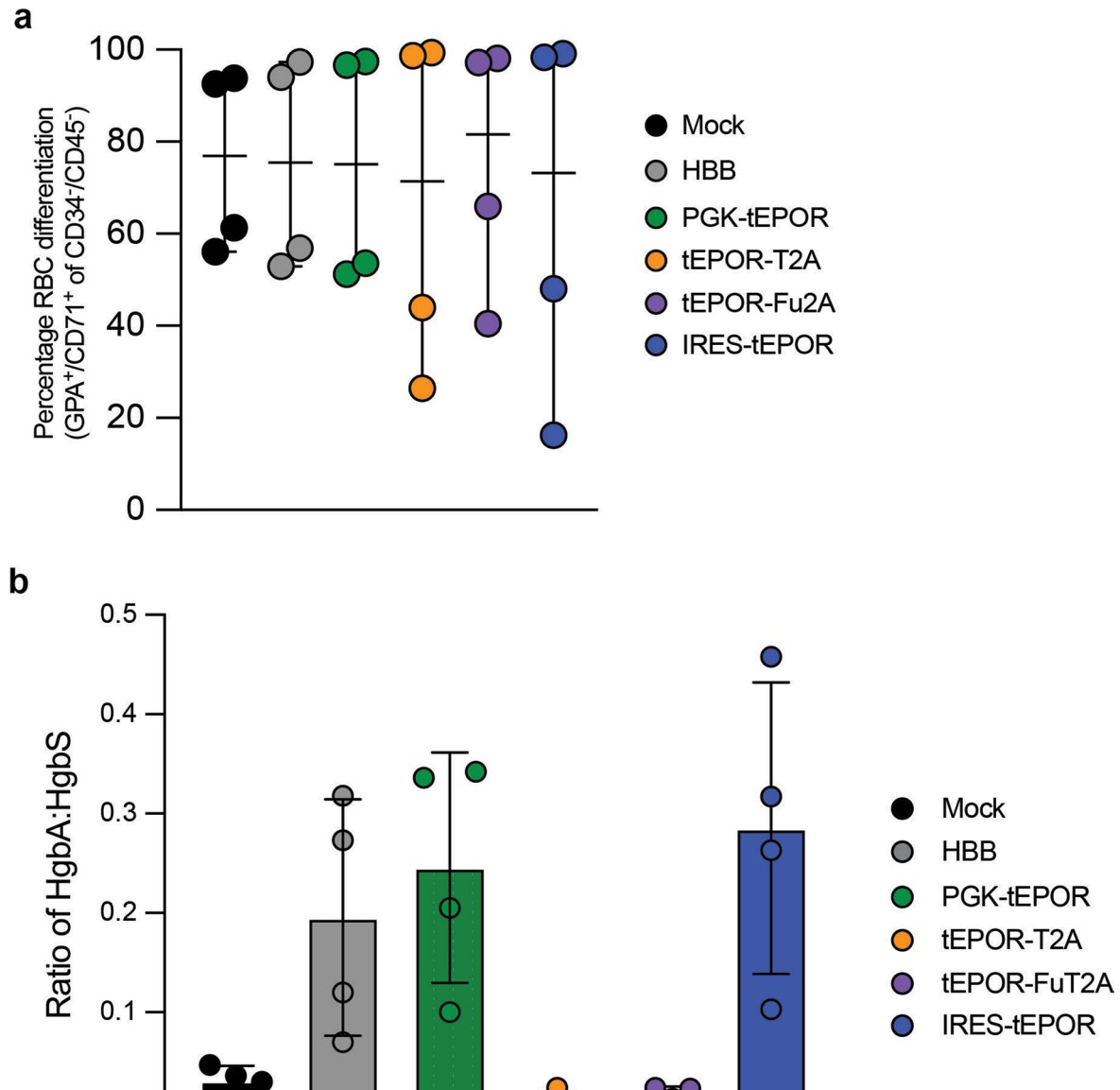

**Extended Data Fig. 7 | Haemoglobin tetramer analysis of *tEPOR*-edited SCD patient HSPCs following RBC differentiation. a**, Percentage of GPA$^+$/CD71$^+$ of CD34$^-$/CD45$^-$ cells on day 14 of RBC differentiation as determined by flow cytometry. Points shown as median ± 95% confidence interval. N = 4 biologically independent HSPC donors. **b**, Ratio of adult haemoglobin to sickle haemoglobin from HPLC analysis of haemoglobin tetramers from differentiated SCD patient HSPCs. Bars shown as mean ± SD. N = 4 biologically independent HSPC donors.

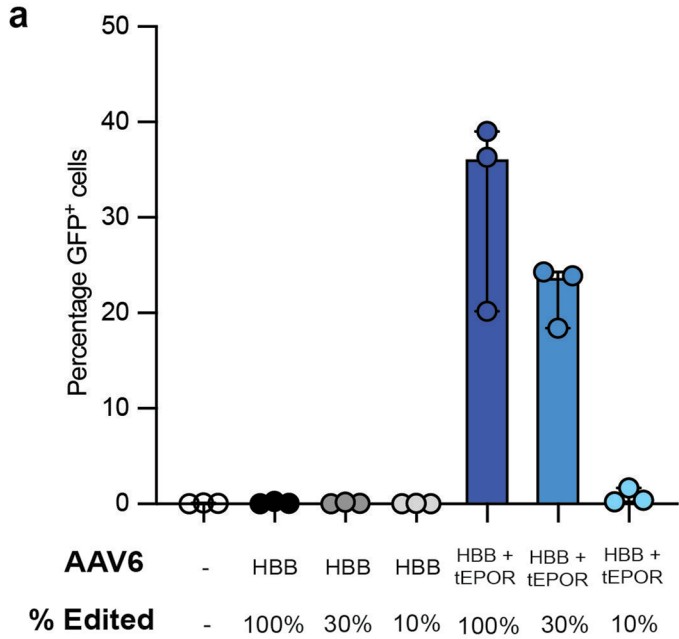

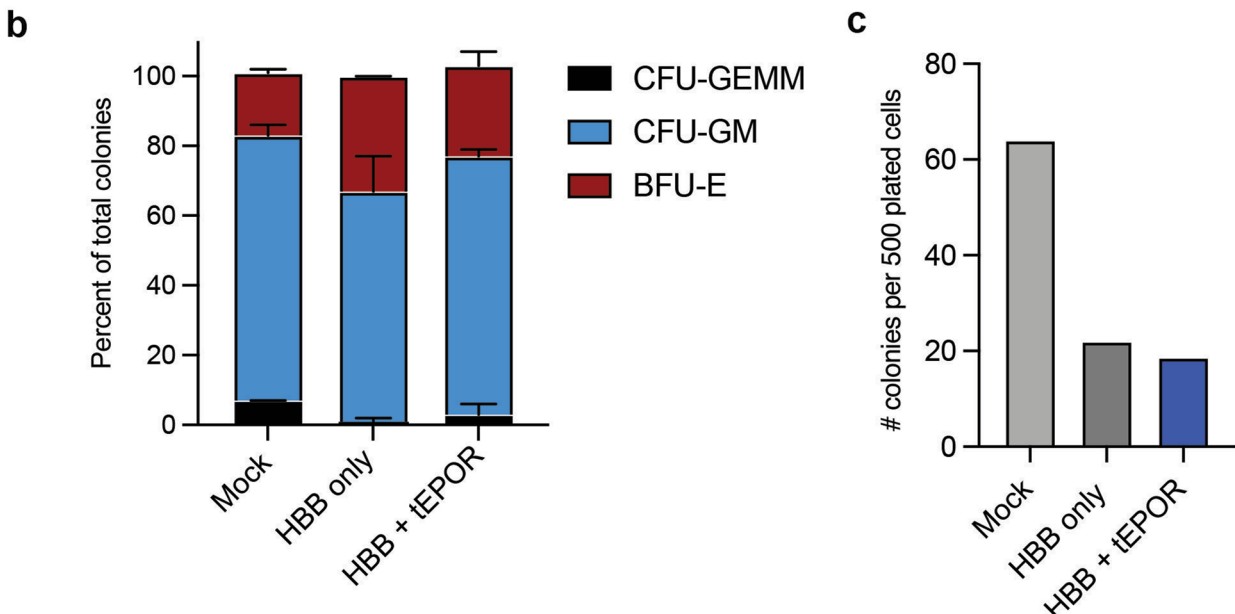

**Extended Data Fig. 8 | Multiplexed editing shows maintenance of *EPOR* truncation at end of RBC differentiation and does not disrupt HSPC lineage formation. a**, Percentage of GFP+ cells of live single cells on day 14 of RBC differentiation as determined by flow cytometry. Points represent median ± interquartile range. Values represent N = 3 biologically independent HSPC donors. **b**, CFU assay of mock, single edited, and multiplex edited HSPCs. Bars represent percent of total colonies: CFU-GEMM (multi-potential granulocyte, erythroid, macrophage, megakaryocyte progenitor cells), CFU-GM (colony forming unit-granulocytes and monocytes), and BFU-E (erythroid burst forming units). N = 1. **c**, Number of total colonies in CFU assay produced per 500 plated cells in each condition.

# Reporting Summary

## Statistics

For all statistical analyses, confirm that the following items are present in the figure legend, table legend, main text, or Methods section.

| n/a | Confirmed | |
|---|---|---|
| ☐ | ☒ | The exact sample size (*n*) for each experimental group/condition, given as a discrete number and unit of measurement |
| ☐ | ☒ | A statement on whether measurements were taken from distinct samples or whether the same sample was measured repeatedly |
| ☐ | ☒ | The statistical test(s) used AND whether they are one- or two-sided<br>*Only common tests should be described solely by name; describe more complex techniques in the Methods section.* |
| ☒ | ☐ | A description of all covariates tested |
| ☒ | ☐ | A description of any assumptions or corrections, such as tests of normality and adjustment for multiple comparisons |
| ☐ | ☒ | A full description of the statistical parameters including central tendency (e.g. means) or other basic estimates (e.g. regression coefficient) AND variation (e.g. standard deviation) or associated estimates of uncertainty (e.g. confidence intervals) |
| ☐ | ☒ | For null hypothesis testing, the test statistic (e.g. *F*, *t*, *r*) with confidence intervals, effect sizes, degrees of freedom and *P* value noted<br>*Give P values as exact values whenever suitable.* |
| ☒ | ☐ | For Bayesian analysis, information on the choice of priors and Markov chain Monte Carlo settings |
| ☒ | ☐ | For hierarchical and complex designs, identification of the appropriate level for tests and full reporting of outcomes |
| ☒ | ☐ | Estimates of effect sizes (e.g. Cohen's *d*, Pearson's *r*), indicating how they were calculated |

*Our web collection on statistics for biologists contains articles on many of the points above.*

## Software and code

Policy information about availability of computer code

| Data collection | Microsoft Excel for Mac (v16.75.2) was used for data collection. ddPCR data were collected using QuantaSoft software (v1.7.4.0917, BioRad). Flow cytometry data were collected through FACS Diva 8 for FACS Aria II. STEMvision automated counter (STEMCELL Technologies) with manual correction was used to to determine the number of BFU-E, CFU-M, CFU-GM and CFU-GEMM colonies following the CFU assay. |
|---|---|
| Data analysis | GraphPad Prism 9 and Microsoft Excel for Mac (v16.75.2) were used for data analysis. INDEL frequency analysis was done through the TIDE analysis online tool (https://tide.nki.nl) on Sanger sequencing samples. FACS data were analysed through FlowJo 10.9.0 for Mac. For off-target analysis, FastQC (v0.11.8, http://www.bioinformatics.babraham.ac.uk/projects/fastqc, default parameters) was employed to assess the quality of raw reads. Subsequently, paired-end reads were aligned to the specified off-target regions using CRISPResso2 (version 2.2.14, CRISPResso --fastq_r1 reads_r1.fastq.gz --fastq_r2 reads_r2.fastq.gz --amplicon_seq). |

For manuscripts utilizing custom algorithms or software that are central to the research but not yet described in published literature, software must be made available to editors and reviewers. We strongly encourage code deposition in a community repository (e.g. GitHub). See the Nature Portfolio guidelines for submitting code & software for further information.

## Data

Policy information about availability of data

All manuscripts must include a data availability statement. This statement should provide the following information, where applicable:

- Accession codes, unique identifiers, or web links for publicly available datasets
- A description of any restrictions on data availability
- For clinical datasets or third party data, please ensure that the statement adheres to our policy

The main data supporting the results in this study are available within the paper and its Supplementary Information. High-throughput-sequencing data generated for off-target analysis is available through the NCBI Sequence Read Archive database via the accession number PRJNA1102034. Sequences of the guide RNA and ddPCR primers and probes are provided in Methods. Source data for the figures are provided with this paper. All data generated in this study are available from the corresponding authors on reasonable request.

## Research involving human participants, their data, or biological material

Policy information about studies with human participants or human data. See also policy information about sex, gender (identity/presentation), and sexual orientation and race, ethnicity and racism.

| | |
|---|---|
| Reporting on sex and gender | Primary cell samples used in this study were de-identified. |
| Reporting on race, ethnicity, or other socially relevant groupings | Primary cell samples used in this study were de-identified. |
| Population characteristics | Primary cell samples used in this study were de-identified. |
| Recruitment | Primary cord-blood and peripheral-blood CD34+ HSPCs were derived from healthy donors at random. HSPCs from patients with sickle cell disease were provided by Vivien A. Sheehan of Emory University (formerly at Baylor College of Medicine). |
| Ethics oversight | The Stanford IRB committee approved the research on healthy donor HSPCs under protocol #33813. Cells from patients with sickle cell disease were collected under the Baylor College of Medicine IRB protocol H-41213. |

Note that full information on the approval of the study protocol must also be provided in the manuscript.

# Field-specific reporting

Please select the one below that is the best fit for your research. If you are not sure, read the appropriate sections before making your selection.

☒ Life sciences    ☐ Behavioural & social sciences    ☐ Ecological, evolutionary & environmental sciences

For a reference copy of the document with all sections, see nature.com/documents/nr-reporting-summary-flat.pdf

# Life sciences study design

All studies must disclose on these points even when the disclosure is negative.

| | |
|---|---|
| Sample size | Experiments were performed in 2–3 biological HSPC donors. |
| Data exclusions | No data were excluded from the analyses. |
| Replication | All attempts at replication were successful. |
| Randomization | The samples were allocated at random. |
| Blinding | There was no blinding in this study. |

# Reporting for specific materials, systems and methods

We require information from authors about some types of materials, experimental systems and methods used in many studies. Here, indicate whether each material, system or method listed is relevant to your study. If you are not sure if a list item applies to your research, read the appropriate section before selecting a response.

## Materials & experimental systems

| n/a | Involved in the study |
|---|---|
| ☐ | ☒ Antibodies |
| ☐ | ☒ Eukaryotic cell lines |
| ☒ | ☐ Palaeontology and archaeology |
| ☒ | ☐ Animals and other organisms |
| ☒ | ☐ Clinical data |
| ☒ | ☐ Dual use research of concern |
| ☒ | ☐ Plants |

## Methods

| n/a | Involved in the study |
|---|---|
| ☒ | ☐ ChIP-seq |
| ☐ | ☒ Flow cytometry |
| ☒ | ☐ MRI-based neuroimaging |

## Antibodies

| | |
|---|---|
| Antibodies used | APC anti-CD34 (BioLegend, cat: 343510, Clone 581, Lot B377029), V450 anti-CD45 (BD Biosciences cat: 560367 Clone: HI30, Lot 1249199), PE-Cy7 anti-CD71 (eBioscience cat: 25071942, Clone: OKT9, Lot 2450617), PE anti-CD235a (eBioscience cat: 12998782, Clone: HIR2, Lot 2450518), Ghost Dye Red 780 (Tonbo Biosciences) |
| Validation | APC anti-CD34 antibody has been validated for flow cytometry in human peripheral blood leukocytes and cited in 18 publications according to the manufacturer's website. V450 anti-CD45 antibody has been validated for flow cytometry in human lymphocytes according to the manufacturer's website. PE-Cy7 anti-CD71 antibody has been validated for flow cytometry in human peripheral blood cells and cited in 23 publications according to the manufacturer's website. PE anti-CD235a antibody has been validated for flow cytometry in human peripheral blood cells and cited in 36 publications according to the manufacturer's website. |

## Eukaryotic cell lines

Policy information about cell lines and Sex and Gender in Research

| | |
|---|---|
| Cell line source(s) | HEK-293T cells were acquired from ATCC. |
| Authentication | HEK-293T cells were authenticated by STR profiling at ATCC. |
| Mycoplasma contamination | The cell line was tested for mycoplasma by the manufacturer, and was found to be negative. |
| Commonly misidentified lines (See ICLAC register) | No commonly misidentified cell lines were used. |

## Flow Cytometry

### Plots

Confirm that:

☒ The axis labels state the marker and fluorochrome used (e.g. CD4-FITC).

☒ The axis scales are clearly visible. Include numbers along axes only for bottom left plot of group (a 'group' is an analysis of identical markers).

☒ All plots are contour plots with outliers or pseudocolor plots.

☒ A numerical value for number of cells or percentage (with statistics) is provided.

### Methodology

| | |
|---|---|
| Sample preparation | Cells were washed once with FACS buffer and then incubated with 4-antibody cocktail (APC anti-CD34, V450 anti-CD45, PE-Cy7 anti-CD71 and PE anti-CD235a) for 15–30 minutes at 4 degrees Celsius. The cells were then washed with FACS buffer. Cells were resuspended in fresh buffer and stained with Ghost Dye Red 780 Live/Dead for at least 5 minutes before analysis. |
| Instrument | FACS Aria II cytometer (BD Biosciences). |
| Software | FACS Diva software 8 was used for data collection. FACS data analysis was performed using FlowJo (10.9.0 for Mac) software. |
| Cell population abundance | n/a |
| Gating strategy | Cells were first gated by FSC-A and Live/Dead. Doublet discrimination was performed using FSC-H/FSC-W and SSC-H/SSC-W. plots. CD34– and CD45– cells were then gated followed by CD71+/CD235a (GPA)+. GFP+ cells were gated by comparing to Mock-edited cells. |

☒ Tick this box to confirm that a figure exemplifying the gating strategy is provided in the Supplementary Information.

