## [Peer Review File · Nature Biomedical Engineering]

Enhancement of erythropoietic output by Cas9-mediated insertion of a natural variant in haematopoietic stem and progenitor cells

Corresponding author: M. Kyle Cromer

Editorial note

This document includes relevant written communications between the manuscript's corresponding author and the editor and reviewers of the manuscript during peer review. It includes decision letters relaying any editorial points and peer-review reports, and the authors' replies to these (under 'Rebuttal' headings). The editorial decisions are signed by the manuscript's handling editor, yet the editorial team and ultimately the journal's Chief Editor share responsibility for all decisions.

Any relevant documents attached to the decision letters are referred to as **Appendix #**, and can be found appended to this document. Any information deemed confidential has been redacted or removed. Earlier versions of the manuscript are not published, yet the originally submitted version may be available as a preprint. Because of editorial edits and changes during peer review, the published title of the paper and the title mentioned in below correspondence may differ.

Correspondence

Wed 15 Nov 2023

Decision on Article nBME-23-2230-T

Dear Dr Cromer,

Thank you again for submitting to *Nature Biomedical Engineering* your manuscript, "Using human genetics to develop strategies to increase erythropoietic output from genome-edited hematopoietic stem and progenitor cells", and for your patience in waiting for our decision. As noted in previous correspondence, we were hoping to receive a third reviewer report. However, the reviewer has now informed me that, unfortunately, owing to personal circumstances, they won't be able to provide the feedback.

Nevertheless, I am satisfied with the quality of the two reviewer reports that I had already forwarded to you and that I have also included at the end of this message. The reviewers appreciate the work, and raise a number of technical criticisms that I am sure you will be able to address. In addition, because the strategy of conferring red blood cells with selective advantage during differentiation has long been explored (although not in the context of CRISPR-based gene editing and using patient-derived cells), please do clearly discuss the earlier efforts directly in the manuscript's introductory text discussing background work. In particular, please discuss the use of lentiviral gene transfer to encode the truncated erythropoietin receptor, as shown in this work published in 2011, and in ref. 30 in your manuscript.

When you are ready to resubmit your manuscript, please upload the revised files, a point-by-point rebuttal to the comments from all reviewers, the reporting summary, and a cover letter that explains the main improvements included in the revision and responds to any points highlighted in this decision.

Please follow the following recommendations:

* Clearly highlight any amendments to the text and figures to help the reviewers and editors find and understand the changes (yet keep in mind that excessive marking can hinder readability).- * If you and your co-authors disagree with a criticism, provide the arguments to the reviewer (optionally, indicate the relevant points in the cover letter).
- * If a criticism or suggestion is not addressed, please indicate so in the rebuttal to the reviewer comments and explain the reason(s).
- * Consider including responses to any criticisms raised by more than one reviewer at the beginning of the rebuttal, in a section addressed to all reviewers.
- * The rebuttal should include the reviewer comments in point-by-point format (please note that we provide all reviewers will the reports as they appear at the end of this message).
- * Provide the rebuttal to the reviewer comments and the cover letter as separate files.

We hope that you will be able to resubmit the manuscript within 10 weeks from the receipt of this message. If this is the case, you will be protected against potential scooping. Otherwise, we will be happy to consider a revised manuscript as long as the significance of the work is not compromised by work published elsewhere or accepted for publication at *Nature Biomedical Engineering*.

We hope that you will find the referee reports helpful when revising the work, which we look forward to receive. Please do not hesitate to contact me should you have any questions.

Best wishes,

Pep

Pep Pàmies
Chief Editor, Nature Biomedical Engineering

Reviewer #1 (Report for the authors (Required)):

The authors of “Using human genetics to develop strategies to increase erythropoietic output from genome-edited hematopoietic stem and progenitor cells” describe the combinatorial use of a correction strategy for beta-hemoglobinopathies together with generating a mutation in the erythropoietin receptor (EPOR) that results in congenital erythrocytosis in humans. The latter edit provides erythroid cells that are corrected for beta-hemoglobinopathies with a selective advantage, which would increase the % of erythroid cells with the desired clinically relevant edits over time.

Thought the findings are interesting and important, a very similar strategy, utilizing the exact same concept, was already published by Uchida et al. in *Sci Transl Med* in 2021, which is referred very briefly in this study (reference 30). The manuscript by Uchida et al utilized the same concept and included in vitro work as well as transplantation experiments in humanized mice and in rhesus macaque. Therefore, my enthusiasm for this work is dampened, because it is not conceptually novel. Technically, the work is robust, and the manuscript is well written.

Major concerns:

1. In figure 3A and extended data figure 7, the authors compare 2A peptide to IRES. However, it appears based on Figure 3A that the HBB is downstream of 2A while it is upstream of the IRES. This makes the comparison between 2A and IRES uninterpretable.
2. The authors mention on several occasions the use of human genetics to develop novel strategies to treat disease. However, this generalization seems overstated, since this manuscript just examines a simple example that has been studied before, as summarized above (Uchida et al, *Sci Transl Med* 2021).

Minor concerns:

3. The authors mention the EPOR mutation occurring in congenital erythropoiesis. The way this is worded suggests that the etiology of congenital erythropoiesis is exclusively from EPOR mutations, while congenital erythropoiesis may result from mutations in a variety of genes.

4. In the second paragraph of the first page of "main" text, the authors mention that the EPOR mutation is benign trait. Though this mutation does not increase the risk of leukemia, I am not entirely certain this it is completely benign, as there may be increased risk of thrombosis with erythrocytosis. Though the latter is not clear in the literature, but it may be more prudent to tone this down a little bit.

5. In line 147-148, the authors estimate that almost all RBCs are derived from cells with either an INDEL or UBC-GFP knock-in. These calculations need to be clarified and explained further.

6. There is inconsistency in the text/figure legends/figure about the use of GFP versus YFP markers. This needs to be clarified.

7. In figure 3C, HBB control should be included.

8. Limitations of the findings are not adequately discussed in the discussion section. For example, the authors try to make the point that the strategy applied may help in situations where the % edit is small. However, in figure 4C, using unedited cell spiking at the start of erythroid differentiation demonstrated that if edited cells consist of 10% of the total cell population, the % of targeted alleles at the end of differentiation is ~6%, which would not be clinically relevant for most hematologic conditions.

Reviewer #2 (Report for the authors (Required)):

Camarena et al. describe strategies to increase erythropoietic output from genome-edited HSPCs by combining a non-pathogenic mutation truncating erythropoietin receptor (tEPOR) together with a gene correction strategy developed for b-thalassemia (HBB mutations).

The authors first utilize CRISPR/Cas9 to create pre-mature stop codons that truncate erythropoietin receptor mimicking the natural tEPOR variant, and precisely knock-in the tEPOR in various loci in HSPCs. They show that the tEPOR enhances erythroid proliferation and hence increases RBC production in vitro differentiation. The authors next develop a strategy to over-express both HBB and tEPOR cDNAs by knocking in various HBB-tEPOR bicistronic cassettes into the human HBA1 locus in healthy donor (HD) or sickle cell disease (SCD) patient-derived HSPCs. With this strategy, the authors show an increase in edited RBC formation. Finally, the authors develop a therapeutic editing strategy by performing multiplexed knock-ins to introduce the tEPOR and HBB correct sequence into the EPOR and HBA1 loci, respectively. In order to increase dual knock-in efficiencies, the authors utilize a DNA-PKcs inhibitor. This strategy also leads to a robust increase in HBB production in edited RBCs.

Overall, this study provides interesting strategies that could be potentially translated into clinical applications to cure RBC-associated diseases. The paper is well written and the study is technically well done and largely quite clear and straightforward.

Major concerns

1. The authors show the benefits of the tEPOR in erythroid proliferation in comparison to non-edited cells in the absence of EPO cytokine. In the presence of EPO (3 U/ml), the RBC production is equal in both unedited and edited cells. What are the effects of the tEPOR in terms of erythroid proliferation in low and high doses of EPO cytokine?

2. In this study, the authors use CRISPR/Cas9 gene editing to knock out the EPOR locus and knock in the tEPOR mutation into various loci, including the endogenous locus. In the abstract, the authors also state "...editing to enable SAFER and..." (line 36). With regard to the safety of gene editing CRISPR/Cas9, the authors should include off-target analysis.

3. In the multiplexed knock-in strategy, the authors use a DNA-PKcs inhibitor that blocks the NHEJ pathway in order to enhance HDR efficiency. How do HSPC growth and yield compare with non-inhibitor conditions?

Minor concerns:

1. In Fig. 2, panel b shows 20-40% of RBCs in “CCR5-sg3 + tEPOR – w/o EPO”, indicating a selective advantage of the tEPOR variant in RBC differentiation. As shown in panel c, > 56% of RBCs express GFP, assuming that total GFP+ cells at the endpoint (day 14) are in a range of 12-24%, however, the data in panel d does not correspond. The authors need to clarify this issue.
2. Fig. 4 shows the data of the multiplexed knock-in strategy, could the authors show the percentages of dual knock-in events in the presence and absence of the DNA-PKcs inhibitor?

Thu 04 Apr 2024

Decision on Article NBME-23-2230A

Dear Dr Cromer,

Thank you for your patience in waiting for the feedback on your revised manuscript, "Using human genetics to develop strategies to increase erythropoietic output from genome-edited hematopoietic stem and progenitor cells". Having consulted with the original reviewers, I am pleased to write that we shall be happy to publish the manuscript in *Nature Biomedical Engineering*, provided that the points specified in the attached instructions file are addressed.

When you are ready to submit the final version of your manuscript, please upload the files specified in the instructions file.

We encourage authors to take up transparent peer review. If you are eligible and opt in to transparent peer review, we will publish, as a single supplementary file, all the reviewer comments for all the versions of the manuscript, your rebuttal letters, and the editorial decision letters. **If you opt in to transparent peer review, in the attached file please tick the box 'I wish to participate in transparent peer review'; if you prefer not to, please tick 'I do NOT wish to participate in transparent peer review'**. In the interest of confidentiality, we allow redactions to the rebuttal letters and to the reviewer comments. If you are concerned about the release of confidential data, please indicate what specific information you would like to have removed; we cannot incorporate redactions for any other reasons. More information on transparent peer review is available.

Best wishes,

Pep

Pep Pàmies
Chief Editor, Nature Biomedical Engineering

P.S. Nature Portfolio journals encourage authors to share their step-by-step experimental protocols on a protocol-sharing platform of their choice. Nature Portfolio's Protocol Exchange is a free-to-use and open resource for protocols; protocols deposited in Protocol Exchange are citable and can be linked from the published article. More details can be found at www.nature.com/protocolexchange/about.

Reviewer #1 (Report for the authors (Required)):

The reviewers have addressed the questions posed. The article is scientifically sound. I have no additional comments or concerns about the scientific aspect of the manuscript.

Reviewer #2 (Report for the authors (Required)):

In the revised manuscript, Luna et al. have addressed all my comments related to their paper. I am satisfied with their clarification and have no further comments. I highly recommend that Nature Biomedical Engineering should publish this study.

Rebuttal 1

Reviewer #1

The authors of “Using human genetics to develop strategies to increase erythropoietic output from genome-edited hematopoietic stem and progenitor cells” describe the combinatorial use of a correction strategy for beta-hemoglobinopathies together with generating a mutation in the erythropoietin receptor (EPOR) that results in congenital erythrocytosis in humans. The latter edit provides erythroid cells that are corrected for beta-hemoglobinopathies with a selective advantage, which would increase the % of erythroid cells with the desired clinically relevant edits over time.

Thought the findings are interesting and important, a very similar strategy, utilizing the exact same concept, was already published by Uchida et al. in *Sci Transl Med* in 2021, which is referred very briefly in this study (reference 30). The manuscript by Uchida et al utilized the same concept and included in vitro work as well as transplantation experiments in humanized mice and in rhesus macaque. Therefore, my enthusiasm for this work is dampened, because it is not conceptually novel. Technically, the work is robust, and the manuscript is well written.

We appreciate the thoughtful feedback and suggestions for improvement. As noted by the reviewer, we referenced the Uchida, *et al.* work in our original submission. We also reference the work that preceded the Uchida, *et al.* work by Oliver Smithies group from nearly two decades ago. Both groups used viral vectors to integrate the tEPOR transgene in a semi-random fashion in the genome. We believe that using genome editing to either directly create the tEPOR variant at the endogenous locus (and in fact, creating the exact variant described in one use case) or by inserting the tEPOR precisely in the genome using targeted integration provides a significant advance in biomedical engineering. We summarize the key advances compared to the Uchida, *et al.* work here:

1. By using genome editing, rather than using a lentiviral vector to insert the transgene under the control of RBC specific promoter, we achieve more precision in harnessing the biology of the truncation. When we engineer the truncation at the endogenous locus, we precisely recreate the human variant that has been found to be safe and not cause pathologic polycythemia. While Uchida, *et al.* show that expression of tEPOR can increase RBC production from modified HSCs, their work hinged on lentiviral-mediated delivery. As a consequence, they were unable to modify cells in a way that replicates the naturally occurring *EPOR* variant.
2. Significant and innovative science is still fundamentally built on the work of others. The precision and versatility of genome editing is a more advanced form of genome engineering than lentiviral strategies for a variety of reasons. One example is that in contrast to Lyfgenia, a lentiviral-based drug for sickle cell disease, Casgevy, a genome editing based drug for sickle cell disease, did not receive a black box warning because of the risk of insertional oncogenesis (which has occurred in other lentiviral-based

gene therapy trials (such as for ALD)). We believe using improved genetic engineering is an important part of the scope for Nature BME.

3. A third advance is that genome editing of allogeneic HSPCs is now in clinical trials with positive engraftment results (see ASH 2023 presentation from Vor Biopharma: <https://www.vorbio.com/publication/ash-oral-presentation-483/>) whereas lentiviral engineering has not been applied to the allo-HSCT setting because of its risks. By showing that we can edit the endogenous locus in HSPCs and give the derived cells a selective advantage we raise the possibility in the discussion of using editing to give allogeneic HSPC increased potency in generating wild-type RBCs. Why would one want to do this? Because it gives a potential strategy to lower the amount of myeloablative conditioning needed to achieve sufficient RBC output to treat the underlying hemoglobinopathy. This approach may synergize with complementary efforts to developing antibody or CAR-T based conditioning and reduce the amount of myeloablative chemotherapy needed. In the allogeneic setting, immunosuppression would still be needed but the short and long-term effects of immunosuppressive agents are more tolerable than the cytotoxic effects of myeloablative agents.

In response to the reviewer's comment and to place our work in the context of prior studies deploying viral-mediated delivery and expression of tEPOR, we have added the following paragraph to the Introduction (Lines 65-76):

“While prior studies have investigated the effects of viral-mediated delivery and expression of *tEPOR* (Uchida, *et al.* Science Translational Medicine, 2021, Negre, *et al.* Blood, 2011, Kirby, *et al.* PNAS, 1996), random insertion into the genomes of billions of hematopoietic stem and progenitor cells (HSPCs) in the context of bone marrow transplant presents a serious safety concern, and has resulted in a “Black Box” warning in the United States for lovetibeglogene autotemcel, a lentiviral gene therapy drug approved for sickle cell disease (Hacein-Bey-Abin, *et al.*, Journal of Clinical Investigation, 2008). In addition, such instances of viral-mediated delivery require expression of tEPOR using a non-native, exogenous promoter, which departs from native EPOR regulation and has the potential for unintended consequences such as pathogenic polycythemia. Nonetheless, viral-mediated expression studies provide a proof-of-concept that demonstrates that tEPOR expression can lend a selective advantage to transduced cells and provide a foundation for the utilization of more advanced genome engineering modalities.”

Major concerns:

1. In figure 3A and extended data figure 7, the authors compare 2A peptide to IRES. However, it appears based on Figure 3A that the HBB is downstream of 2A while it is

upstream of the IRES. This makes the comparison between 2A and IRES uninterpretable.

We thank the reviewer for their astute observation regarding the positioning of *HBB* in relation to 2A and IRES in Figure 3A and Extended Data Figure 12 (formerly Extended Data Figure 7) and recognize the implications it may have on the interpretability of the comparison. The positioning of *HBB* downstream of 2A is based on our unpublished data from a previously published work (Cromer, *et al.* Nature Medicine, 2021). In that study, we observed that when *HBB* is upstream of the 2A, there is a marked absence of adult hemoglobin tetramer (HgbA) production within a YFP⁺-sorted population integrated with *HBB*-T2A-YFP vector (see figure below; orange arrow compared to blue arrow when edited with an identical vector without 2A-YFP). We hypothesized that the presence of a longer cleavage tail on the C-terminus of the upstream HBB protein is likely interfering with formation of the hemoglobin tetramer complex. This is in line with previous findings (Reinhardt, *et al.*, Stem Cell Rep. 2020) that reported disruption of protein function due to 2A peptide cleavage tails. Because 2A peptides leave a much shorter (1 proline) cleavage tail on the N-terminus of the downstream protein (Liu, *et al.* Sci Rep, 2017), in this study we chose to exclusively test *HBB* downstream of the 2A peptide. The observed constraints of the 2A peptide also prompted our exploration of Furin-T2A and IRES as a potentially more effective strategy that still maintained therapeutic *HBB* to be under endogenous *HBA1* expression. As such, our comparison of these two vectors sought to define the most effective strategy for optimal protein production via HPLC, with our findings favoring IRES in this instance.

Unpublished data that C-terminal 2A peptide cleavage tail disrupts hemoglobin tetramer formation

Figure legend: Absence of adult hemoglobin production—abbreviated “A” in the sorted YFP⁺ population, whereas SCD HSPCs edited with the *HBB* only vector show a distinct increase in adult hemoglobin; citation below reports disruption of protein becoming part of the hemoglobin tetramer due to 2A peptide cleavage tail.

Citations:

1. Reinhardt A, Kagawa H, Woltjen K. N-Terminal Amino Acids Determine KLF4 Protein Stability in 2A Peptide-Linked Polycistronic Reprogramming Constructs. *Stem Cell Rep.* 2020;14(3):520-527. doi:10.1016/j.stemcr.2020.01.014

2. Liu Z, Chen O, Wall JBJ, et al. Systematic comparison of 2A peptides for cloning multi-genes in a polycistronic vector. *Sci Rep.* 2017;7(1):2193. doi:10.1038/s41598-017-02460-2

2. The authors mention on several occasions the use of human genetics to develop novel strategies to treat disease. However, this generalization seems overstated, since this manuscript just examines a simple example that has been studied before, as summarized above (Uchida et al, *Sci Transl Med* 2021).

The reviewer is correct that we only investigated the expression of a single naturally occurring variant to increase erythropoietic output from genome-edited HSPCs. The work

of Uchida, *et al.* and the Smithies group, however, did not actually create the natural human variant of the tEPOR as we did here. In terms of other examples, we add this in the first paragraph of the manuscript to the increasing long list in which genome editing has been used to re-create natural human genetic variants for therapeutic purposes (CCR5, PCSK9, HPFH). We expect that as the physiology of both rare and common human genetic variation continues to be understood, genome editing will be a mechanism to apply those learnings in the therapeutic setting. We put this work into that bucket in which human genetics, precision genome engineering, and understanding of human disease pathophysiology are melded into an innovative piece of work.

Minor concerns:

3. The authors mention the EPOR mutation occurring in congenital erythropoiesis. The way this is worded suggests that the etiology of congenital erythropoiesis is exclusively from EPOR mutations, while congenital erythropoiesis may result from mutations in a variety of genes.

In response to the reviewer's insightful comment, we have revised the pertinent sections of the abstract and introduction to reflect a more accurate representation of the role of *EPOR* mutations in congenital erythrocytosis that are summarized below.

Abstract

Line 28 - 31

One of the best characterized naturally occurring mutations causing congenital erythrocytosis arises from a truncation in the erythropoietin receptor (tEPOR) which can result in non-pathogenic hyper-production of red blood cells (RBCs).

Introduction

Line 49 - 55

Congenital erythrocytosis (CE) is a rare phenotype in which people have higher than normal levels of red blood cells and consequently elevated hemoglobin. While there are multiple genetic variants that can lead to this condition, perhaps the best characterized genotype was first identified in the family of a Finnish Olympic gold medal-winning cross-country skier who was found to have levels of hemoglobin >50% higher than normal⁴. This was attributed to truncations in the erythropoietin receptor (tEPOR) in which the intracellular inhibitory domain to erythropoietin (EPO) signaling is eliminated^{4,5}.

4. In the second paragraph of the first page of “main” text, the authors mention that the EPOR mutation is benign trait. Though this mutation does not increase the risk of leukemia, I am not entirely certain this it is completely benign, as there may be increased

risk of thrombosis with erythrocytosis. Though the latter is not clear in the literature, but it may be more prudent to tone this down a little bit.

In response to the reviewer's observation, we have modified the section from lines 57-64 to provide a more nuanced description regarding the nature of the *EPOR* mutation.

Line 57-64:

Further studies have shown that tEPOR does not create a constitutively active EPOR signaling cascade, but rather imparts hypersensitivity to EPO^{5,6}. As a consequence, these kindreds with tEPOR typically present with abnormally low levels of EPO, indicating a new homeostasis is attained to prevent CE from becoming pathogenic. There have been reports of thrombotic and hemorrhagic events likely due to erythrocytosis but many of these have a benign clinical course⁷. More importantly, families with CE have not shown an increased predisposition to cancer, demonstrating that this is not a pre-malignant genetic condition⁸.

5. In line 147-148, the authors estimate that almost all RBCs are derived from cells with either an INDEL or UBC-GFP knock-in. These calculations need to be clarified and explained further.

We thank the reviewer for this helpful comment. To clarify our calculations, we conducted an analysis of INDELS on three biological replicates where cells were targeted with the tEPOR-Ubc-GFP vector. On day 14 differentiated RBCs in three donors, we see a median 51.1% GFP⁺ cells and median 92.1% indels, via flow cytometry and TIDE analysis, respectively. We have added this data to Extended Data Figure 4a (shown below). This quantification should better clarify the estimation of total RBCs with either a truncating indel or an integration event. This data demonstrates that 51% of alleles have a targeted insertion event creating the natural variant and 92% of the remaining alleles have an INDEL which also functionally leads to tEPOR. In Extended Data Figure 4a, one can see that both INDELS and GFP⁺ cells enriched over time (as expected) as both should compete equivalently against unmodified wild-type alleles.

Extended Data Figure 4

6. There is inconsistency in the text/figure legends/figure about the use of GFP versus YFP markers. This needs to be clarified.

Thank you for this note. We have updated the text and figures to correct these discrepancies. In Figure 1, GFP was used while in Figure 2, YFP was used.

Corrections have been made to the following:

- Figure 2c flow cytometry plot label
- Figure 2d Panel title and axis label
- Figure 2g flow cytometry label
- Figure 2h Panel title and axis label
- Figure 2d, h Legend
- Line 269

7. In figure 3C, HBB control should be included.

We have added data for the *HBB* control in healthy CD34⁺ cells in Figure 3c.

8. Limitations of the findings are not adequately discussed in the discussion section. For example, the authors try to make the point that the strategy applied may help in situations where the % edit is small. However, in figure 4C, using unedited cell spiking at the start of erythroid differentiation demonstrated that if edited cells consist of 10% of the total cell population, the % of targeted alleles at the end of differentiation is ~6%, which would not be clinically relevant for most hematologic conditions.

We appreciate the reviewer's comment regarding the "spike-in" experiments, which we used to mimic the context of edited HSCs competing with unedited HSCs in the bone marrow post-transplantation. Since the initial editing frequency was 20-30%, in the 100% experiment only 20-30% of the cells are edited, in the "30%" condition only 6-10% of the

population is edited ($30\% \times 0.2-0.3$), and in the “10%” condition only 2-3% of the cells are edited ($10\% \times 0.2-0.3$).

Our intention here was to understand the degree of selective advantage imparted to edited cells during medium and low levels of engraftment, and whether lower levels of engraftment might generate an even greater selective advantage. Indeed, we observed that at the 30% spike-in condition, we see approximately a 2-fold increase in cells co-expressing *HBB*+*tEPOR* compared to cells edited with *HBB* alone. As expected, we also see an even greater selective advantage (~3-fold) at 10% spike-in of dual-edited cells compared to *HBB*-edited cells. While the reviewer is correct to say that 6% edited alleles in the instance of 10% “engraftment” would not be clinically corrective, the goal of these experiments was to push the limits of our system to understand if *tEPOR* could impart an even greater selective advantage if low engraftment were to occur. We also point out that in some forms of beta-thalassemia, such as beta+ thalassemia or HbE disease, a 3-fold selective enrichment might transform a patient from transfusion-dependent to transfusion-independent without having to achieve high engraftment frequencies.

Reviewer #2:

Camarena et al. describe strategies to increase erythropoietic output from genome-edited HSPCs by combining a non-pathogenic mutation truncating erythropoietin receptor (*tEPOR*) together with a gene correction strategy developed for β -thalassemia (*HBB* mutations).

The authors first utilize CRISPR/Cas9 to create pre-mature stop codons that truncate erythropoietin receptor mimicking the natural *tEPOR* variant, and precisely knock-in the *tEPOR* in various loci in HSPCs. They show that the *tEPOR* enhances erythroid proliferation and hence increases RBC production in vitro differentiation. The authors next develop a strategy to over-express both *HBB* and *tEPOR* cDNAs by knocking in various *HBB*-*tEPOR* bicistronic cassettes into the human *HBA1* locus in healthy donor (HD) or sickle cell disease (SCD) patient-derived HSPCs. With this strategy, the authors show an increase in edited RBC formation. Finally, the authors develop a therapeutic editing strategy by performing multiplexed knock-ins to introduce the *tEPOR* and *HBB* correct sequence into the *EPOR* and *HBA1* loci, respectively. In order to increase dual knock-in efficiencies, the authors utilize a DNA-PKcs inhibitor. This strategy also leads to a robust increase in *HBB* production in edited RBCs.

Overall, this study provides interesting strategies that could be potentially translated into clinical applications to cure RBC-associated diseases. The paper is well written and the study is technically well done and largely quite clear and straightforward.

Major concerns

1. The authors show the benefits of the tEPOR in erythroid proliferation in comparison to non-edited cells in the absence of EPO cytokine. In the presence of EPO (3 U/ml), the RBC production is equal in both unedited and edited cells. What are the effects of the tEPOR in terms of erythroid proliferation in low and high doses of EPO cytokine?

We appreciate the note on the role of tEPOR in erythroid proliferation across varying EPO concentrations. To address this, we have edited HSPCs with tEPOR targeting *EPOR* (truncating the endogenous gene), *CCR5* (expressing tEPOR cDNA from UbC promoter), and *HBA1* (expressing tEPOR cDNA from *HBA1* promoter) with subsequent RBC differentiation assessed across a gradient of EPO concentrations in the media: 0, 0.3, 1, 3, and 20 U/mL which is summarized below and also included in Extended Data Figure 5a-d, 8c-e, and 10b-d. Overall, we found little impact at the lower and higher concentrations of EPO tested, indicating that tEPOR could sensitize cells to EPO even at 3-10-fold lower concentrations than that initially tested (3 U/mL). These results are consistent with the first descriptions of the naturally occurring tEPOR in which there is increased sensitivity to EPO (Sokol, *et al.* Blood 1995; Juvonen *et al.* Blood 1991).

2. In this study, the authors use CRISPR/Cas9 gene editing to knock out the EPOR locus and knock in the tEPOR mutation into various loci, including the endogenous locus. In the abstract, the authors also state "...editing to enable SAFER and..." (line 36). With regard to the safety of gene editing CRISPR/Cas9, the authors should include off-target analysis.

We thank the reviewer for this point, since any novel genome editing strategy needs to be appropriately studied to ensure that unintended edits are not occurring in functionally

important regions of the genome. We therefore performed *in silico* analysis using the COSMID tool to predict candidate off-target sites for the tEPOR sg1 gRNA, which was used throughout this study. We found that 94% (73/78) of candidate off-target sites with COSMID score ≤ 5.5 (shown in prior work cited below that this was most informative for identifying sites with potential off-target activity) resided in intergenic or intronic regions of the genome (see below). This analysis revealed 2 candidate off-target sites in the 3' UTR of genes as well as 3 candidate off-target sites in exonic regions of genes. Because disruption of coding regions would be most concerning from the perspective of safety, we performed targeted amplicon sequencing of these 5 candidate off-target sites residing in coding regions of the genome. We found that all sites were sequenced to high read depth ($>100,000$) and found no evidence of off-target activity when cells were edited with Cas9 protein complexed with *EPOR* sgRNA. These data are summarized below and included in the revised manuscript as Extended Data Figure 7.

Extended Data Figure 7

a Predicted sites with score ≤ 5.5

	Chromosomal Position	Score	Region	Gene
On-target	Chr16:11378177-11378199	0	Exon	EPOR
Off-target	Chr4:2834819-2834841	0.54	5' UTR	SH3BP2
	Chr17:831455508-831455530	0.61	Intergenic	
	Chr14:74552967-74552989	0.63	Exon	LTBP2
	Chr17:76190490-76190511	1.02	Intron	RNF157
	Chr17:74795185-74795187	1.08	Intron	TMEM104
	Chr19:57614583-57614605	1.22	Intron	ZNF134
	Chr22:23681747-23681770	1.25	Intron	GUSBP11
	Chr1:133742338-133742361	1.44	Intergenic	
	Chr1:109777058-109777079	1.48	Intergenic	
	Chr17:3725613-3725635	1.5	Exon	HASPW
	Chr20:47957087-47957110	1.51	Intergenic	
	Chr15:62887187-62887188	1.52	Intergenic	
	Chr1:153353786-153353807	1.53	Intergenic	
	Chr22:47588374-47588397	1.70	Intergenic	
	Chr5:66475949-66475972	1.82	Intergenic	
	Chr15:35375033-35375054	1.83	Intron	DFP16
	Chr3:146684180-146684183	2.08	Intergenic	
	ChrX:53455142-53455183	2.15	Intergenic	
	Chr18:58089238-58089259	2.15	Intergenic	
	Chr15:66356597-66356618	2.15	3' UTR	TPW
	Chr11:89809008-89809031	2.17	Intergenic	
	Chr1:125489832-125489853	2.21	Intergenic	
	Chr1:1741386-1741407	2.35	Intergenic	
	Chr1:74615621-74615642	2.35	Intergenic	
	Chr2:207213231-207213252	2.66	Intron	SLU-AS1
	Chr7:884230583-88423084	2.69	Intergenic	
	Chr1:73358289-73358311	2.72	Intergenic	
	Chr2:45355014-45355035	2.85	Intergenic	
	Chr3:80743788-80743789	2.95	Intron	PHI7
	Chr1:103967054-103967075	3.04	Intergenic	
	Chr3:22888578-22888600	3.06	Intergenic	
	Chr18:88736230-88736251	3.11	Exon	PIEZO1
	Chr2:120715491-120715512	3.11	Intergenic	
	ChrX:68993580-68993602	3.17	Intergenic	
	Chr5:43346327-43346348	3.18	Intergenic	
	Chr1:123359014-123359035	3.2	Intron	CR2
	ChrX:30077883-30077904	3.21	Intergenic	
	Chr22:34432884-34432715	3.21	Intergenic	
	Chr21:45947890-45947882	3.31	Intergenic	
	Chr12:94343003-94243025	3.35	Intron	FLXNC1
	Chr18:7952301-7952323	3.4	Intron	ATP9B
	Chr8:144201447-144201489	3.42	Intron	MROH1
	Chr2:3858090-3859111	3.48	Intergenic	
	Chr5:23614195-23614216	3.50	Intergenic	
	Chr10:44054893-44054715	3.62	Intergenic	
	Chr3:59498371-59498393	3.62	Intergenic	
	Chr22:48855882-48855883	3.63	Intron	LINC01310
	Chr22:236161588-236161609	3.63	Intergenic	
	Chr13:78621038-78621050	3.66	Intergenic	
	ChrX:89384155-89384178	3.78	Intergenic	
	Chr12:22043322-22043343	3.89	Intron	CAC515
	Chr19:54221554-54221575	3.89	Intron	LIR3
	Chr19:54241302-54241323	3.89	Intron	LIRA6
	Chr3:171095217-171095238	3.95	Intron	TNK
	Chr17:29069628-29069649	3.96	Intergenic	
	Chr5:79849804-79849827	4.02	Intergenic	
	Chr13:23853887-23853888	4.03	Intron	TNFRSF19
	Chr13:3225407-3225428	4.03	Intergenic	
	Chr17:3169786-3169807	4.09	Intron	LOC100288728
	Chr1:209455360-209455383	4.12	Intergenic	
	Chr22:42906036-42906057	4.2	Intron	PACSN2
	Chr10:48290891-48290913	4.36	Intergenic	
	Chr1:136644277-136644299	4.41	Intergenic	
	Chr19:57018808-57018830	4.5	Intergenic	
	Chr22:25314961-25314982	4.66	Intergenic	
	Chr20:36849898-36849917	4.8	Intergenic	
	Chr5:1374116-1374137	4.82	Intergenic	
	Chr1:223229923-223229944	4.88	Intron	SUSD4
	Chr5:54602798-54602819	4.95	Intergenic	
	Chr14:103074393-103074414	4.95	Intergenic	
	Chr18:50151538-50151561	4.99	Intergenic	
	Chr7:91541296-91541318	5	Intergenic	
	Chr5:52211164-52211187	5.08	Intergenic	
	Chr5:122639266-122639287	5.14	Intron	LINC02201
	Chr14:105668438-105668459	5.18	Intergenic	
	Chr13:44678373-44678396	5.39	Intron	LINC00407
	Chr1:154810236-154810258	5.4	Intron	KCNK3
	Chr16:47975646-47975668	5.49	Intergenic	

b OT distribution by region in genome

c Exonic and UTR OT Sites

Off-target Site	Chromosomal Position	COSMIC Score	Sequence	Mismatch	Strand	Cut Site	Gene	Region of Gene
OT1	Chr4:2834819-2834841	0.54	TGCTACGGGACAGTGCAGGG - ht AGCTAAGGGACAGTGCAGGG - query	3	+	2834835	SH3BP2	5' UTR
OT2	Chr14:74552967-74552989	0.63	AGCTGTGGGACAGTGCAGGG - ht AGCTAAGGGACAGTGCAGGG - query	3	+	74552983	LTBP2	Exon
OT3	Chr17:3725613-3725635	1.5	AACCTGAGCACTGTCCAGGG - ht AGCTAAGGGACAGTGCAGGG - query	3	+	3725629	HASPW	Exon
OT4	Chr15:66356597-66356618	2.15	AACCTGAGCACTGTCCAGGG - ht AGCTAAGGGACAGTGCAGGG - query	2	+	6635612	TPW	3' UTR
OT5	Chr18:88736230-88736251	3.11	AGCTAAGGGACAGTGCAGGG - ht AGCTAAGGGACAGTGCAGGG - query	1	+	88736245	PIEZO1	Exon

d NGS Read Depth

e Frequency of reads with indels (%)

Citation:

Cromer MK, Majeti KR, Rettig GR, et al. Comparative analysis of CRISPR off-target discovery tools following ex vivo editing of CD34+ hematopoietic stem and progenitor cells. *Molecular Therapy*. 2023;31(4):1074-1087. doi:10.1016/j.ymthe.2023.02.011

3. In the multiplexed knock-in strategy, the authors use a DNA-PKcs inhibitor that blocks the NHEJ pathway in order to enhance HDR efficiency. How do HSPC growth and yield compare with non-inhibitor conditions?

We thank the reviewer for raising this pertinent point regarding the effects of the DNA-PKcs inhibitor on HSPC health. Given the fact that this point has been explored extensively in a publication from our group Selvaraj, *et al.*, *Nat Biotechnology*, 2023 (citation below), we did not perform comparisons between cells edited with and without the DNA-PKcs inhibitor in this study. In this prior work, across a variety of different primary cell types, the authors found no significant effects on cell viability at the optimal concentration of DNA-PKcs inhibitor. In particular, the Extended Data Figure 5c shows that HSPCs edited with RNP+AAV6 with and without DNA-PKcs inhibitor show no differences in the colony forming ability of the HSPCs. Additionally, in Extended Data Figure 5f, no differences in cell viability were observed at three different loci.

Extended Data Figure 5 (Selvaraj, *et al.*, *Nature Biotechnology* 2023):

Citation:

Selvaraj S, Feist WN, Viel S, et al. High-efficiency transgene integration by homology-directed repair in human primary cells using DNA-PKcs inhibition. *Nat Biotechnol*. Published online August 3, 2023:1-14. doi:10.1038/s41587-023-01888-4

Minor concerns:

1. In Fig. 2, panel b shows 20-40% of RBCs in “CCR5-sg3 + tEPOR – w/o EPO”, indicating a selective advantage of the tEPOR variant in RBC differentiation. As shown in panel c,

>56% of RBCs express GFP, assuming that total GFP+ cells at the endpoint (day 14) are in a range of 12-24%, however, the data in panel d does not correspond. The authors need to clarify this issue.

We thank the reviewer for pointing out these potential discrepancies in Figure 2. To clarify the points of confusion highlighted, we have provided clearer explanations in the respective figure legends of what data is being shown in each panel. The plot below from Figure 2c depicts a representative flow cytometry plot from day 14 of RBC differentiation in the +EPO condition to illustrate both CD71⁻/GPA⁻ and CD71⁺/GPA⁺ cells expressing YFP. Given the high frequencies of RBC differentiation by day 14 of RBC differentiation, we chose to display data from a donor which has high frequencies of cells that both have and have not acquired RBC markers.

2. Fig. 4 shows the data of the multiplexed knock-in strategy, could the authors show the percentages of dual knock-in events in the presence and absence of the DNA-PKcs inhibitor?

We thank the reviewer for bringing up this point. In previous work by our group published in Selvaraj, *et al.*, Nature Biotechnology, 2023, it was found that biallelic editing at a single locus increased by more than seven-fold in edited HSPCs treated with the DNA-PKcs inhibitor at the same concentration (0.5 μ M) used in our experiments compared to untreated edited HSPCs. Based on these prior results, we expect the frequency of double knock-in would be significantly lower without using the DNA-PKcs inhibitor and thus the desired coupling of the two events would not reach sufficient frequency to increase the percentage of cells with the *HBA1* knock-in.

Citation:

Selvaraj S, Feist WN, Viel S, et al. High-efficiency transgene integration by homology-directed repair in human primary cells using DNA-PKcs inhibition. *Nat Biotechnol.* Published online August 3, 2023:1-14. doi:[10.1038/s41587-023-01888-4](https://doi.org/10.1038/s41587-023-01888-4).